

# Towards the Assimilation of Atmospheric $CO_2$ Concentration Data in a Land Surface Model using Adjoint-free Variational Methods

Simon Beylat[1,2], Nina Raoult[3], Cédric Bacour[1], Natalie Douglas[4], Tristan Quaife[4], Vladislav Bastrikov[1], Peter J. Rayner[2], and Philippe Peylin[1]

[1]Laboratoire des Sciences du Climat et de l'Environnement, LSCE/IPSL, CEA-CNRS-UVSQ, Université Paris-Saclay, 91191 Gif-sur-Yvette, France
[2]School of Geography, Earth and Atmospheric Sciences, University of Melbourne, Parkville, 3010 Victoria, Australia
[3]European Centre for Medium-Range Weather Forecasts, Shinfield Park, Reading, UK
[4]National Centre for Earth Observation, Department of Meteorology, University of Reading, Reading, UK

**Correspondence:** simon.beylat@lsce.ipsl.fr

**Abstract.**

A comprehensive understanding and an accurate modelling of the terrestrial carbon cycle, are of paramount importance to improve projections of the global carbon cycle and more accurately gauge its impact on global climate systems. Land Surface Models, which have become an important component of weather and climate applications, simulate key aspects of the terrestrial carbon cycle such as photosynthesis and respiration. These models rely on parameterisations that necessitate to be carefully calibrated. In this study we explore the assimilation of atmospheric $CO_2$ concentration data for parameter calibration of the ORCHIDEE Land Surface Model using 4DEnVar, an adjoint-free ensemble-variational data assimilation method. By circumventing the challenges associated with developing and maintaining tangent linear and adjoint models, the 4DEnVar method offers a very promising alternative. Using synthetic observations generated through a twin experiment, we demonstrate the ability of 4DEnVar to assimilate atmospheric $CO_2$ concentration for model parameter calibration. We then compare the results to a 4DVar method that uses finite differences to estimate tangent linear and adjoint models, which reveal that 4DEnVar is superior in terms of computational efficiency and fit to the observations as well as parameter recovery.

## 1 Introduction

Since the link between the increase in atmospheric $CO_2$ concentration and global warming was revealed, understanding the carbon cycle has become essential. This increase is mainly due to anthropogenic emissions (IPCC, 2023), half of which are absorbed by oceans and lands. To improve predictions of the carbon cycle and reduce its associated uncertainty in climate projections, it is essential to better understand the mechanism of the carbon sink, particularly its land component, which remains the most uncertain aspect of the global carbon budget (Friedlingstein et al., 2023).

Atmospheric $CO_2$ concentration data have long been considered a rich source of information to understand the global carbon cycle and characterise the spatio-temporal variation of natural $CO_2$ fluxes (Kaminski et al., 1999a; Rayner et al., 1999; Bousquet et al., 2000; Gurney et al., 2002; Peylin et al., 2005, 2013; Chevallier et al., 2014). Given that the atmosphere



is relatively well mixed, the observed concentration gradients (in space and time) can be used to identify the large-scale characteristics of the underlying surface fluxes. Indeed this surface fluxes are the primary drivers of these gradients. Studying these data gives us an overall view of all the components of the carbon cycle. For more than 25 years, atmospheric $CO_2$

inversions have been used to estimate natural $CO_2$ surface fluxes, using atmospheric transport models and Bayesian inversion frameworks (Kaminski et al., 1999b; Enting, 2002; Chevallier et al., 2005, 2007; Baker et al., 2006; Rayner et al., 2019; Berchet et al., 2021). Atmospheric transport models represent the transport of atmospheric tracers, making it possible to simulate the 3D fields of atmospheric $CO_2$ concentration based on a $CO_2$ surface flux scenario, including all components of the carbon cycle: natural land and ocean fluxes, and anthropogenic emissions from fossil fuels and cement. By inverting the atmospheric

transport and using the $CO_2$ surface flux scenario as prior information, atmospheric inversions statistically adjust the surface $CO_2$ fluxes, minimising the differences between observed and modelled concentrations. This statistical optimisation generally assumes that the corrections to $CO_2$ surface fluxes are isotropic in time and space. While this approach has been valuable for understanding the global carbon cycle, it only estimates the net surface fluxes, with no direct information on the underlying components (i.e. photosynthesis uptake, ecosystem respiration release, fire release, etc.). Consequently, this approach is also

not suitable to make future projections.

Over the same period, land surface models (LSMs) have become an important component of Earth system models, representing a wide range of interactions between the land surface and the atmosphere. As their role has expanded, these models have incorporated an increasing number of complex processes (Fisher and Koven, 2020), and have come to play a key role in weather and climate applications. LSMs now simulate key aspects of the terrestrial carbon cycle, including soils and vegetation

dynamics, providing valuable insights into the main drivers of the land carbon budget and enabling future projections. Given the complexity and the small-scale nature of many of these processes, they are represented using mechanistic and empirical formulations. To accurately model these processes, LSMs rely on parametrisations that must be carefully calibrated to ensure their simulations are consistent with actual observations. One promising approach for calibrating these parameters is the use of atmospheric $CO_2$ concentration data, which offers a global constraint for large-scale calibration, serving as an alternative to

traditional atmospheric inversions (Knorr and Heimann, 1995; Kaminski et al., 2002, 2012; Rayner et al., 2005; Scholze et al., 2007; Peylin et al., 2016; Schürmann et al., 2016; Castro-Morales et al., 2019; Bacour et al., 2023). This assimilation enables the calibration of LSM parameters by adjusting the underlying process representations rather than directly modifying the fluxes themselves. Such an approach also helps to identify structural errors within the models and enhances our understanding of the various processes involved. Once calibrated and refined, these models can be applied to generate more reliable future projec-

tions.

There is a long history of using data assimilation frameworks to calibrate LSM parameters (Rayner, 2010; MacBean et al., 2022; Raoult et al., 2024b). Popular methods - also used in atmospheric inversions - are variational Bayesian methods and, more specifically, the 4DVar method. This method was originally developed in meteorology and Earth sciences to correct

the initial state of the model and has been shown to be robust and very efficient (Courtier et al., 1994; Asch et al., 2016). The 4DVar approach involves defining a cost function (which is usually based on a least-square criterion) that computes the



difference between observations and model outputs as well as a background term that accounts for prior knowledge of the parameters. In order to minimise this cost function, the 4DVar method calculates its gradient with respect to the different parameters to be calibrated. A precise calculation of the gradient of this cost function requires the tangent linear and the adjoint

model (Plessix, 2006). To obtain these models, the code must be differentiated. This task can be performed using automatic differentiation software (Giering and Kaminski, 2003), but the model code must be cleaned up and small modifications made to ensure differentiation (e.g. the reformulation of minimum and maximum computations to enable a smooth transition at the edge, Schürmann et al. (2016)). For some LSMs, it is possible to keep the model compliant using automatic differentiation software (Kaminski et al., 2012; Knorr et al., 2024), however, for complex community models such as ORCHIDEE or JULES

LSMs (Raoult et al., 2016), maintaining the tangent linear and adjoint models is very challenging due to their continuous evolution. In this case, one approach to calculating the gradient is to use finite differences to estimate the gradient of the cost function in order to use the 4DVar method (Santaren et al., 2007; MacBean et al., 2015; Peylin et al., 2016; Bacour et al., 2019).

Several avenues of research have been explored for parameter calibration, including alternative methods to minimise the cost function and the application of new machine learning techniques (Raoult et al., 2024a, b). Ensemble methods have proven

effective for the calibration of LSM parameters, such as Genetic Algorithm (GA) (Santaren et al., 2014; Bastrikov et al., 2018) or Markov chain Monte Carlo (MCMC) (Ziehn et al., 2012). These methods require a large number of simulations and are primarily used with low-cost computational models and for on-site applications, as here they are relatively inexpensive. Parameter calibration in Earth system models has also been the subject of more intensive research (Hourdin et al., 2017). It has led to the development of new methods - emulator-based methods (Williamson et al., 2013; Couvreux et al., 2021) for instance - that have

been used to calibrate components of Earth system models (Watson-Parris et al., 2021; Hourdin et al., 2023) such as ocean and atmospheric model (Williamson et al., 2017; Hourdin et al., 2021; King et al., 2024). In these methods, the model is replaced by an emulator - a computationally efficient statistical model designed to reproduce the behaviour of complex models - to enable numerous simulations and rule out sets of parameters that are not plausible. These methods are gaining in popularity for the calibration of LSM parameters (Dagon et al., 2020; Baker et al., 2022; McNeall et al., 2024; Raoult et al., 2024a) but they

still require a large ensemble of simulations to build the emulator. More recently, an ensemble 4DVar method named 4DEnVar implemented in (Pinnington et al., 2020) for LSM parameter estimation has proved very promising. This method uses a small ensemble to circumvent the necessity for a tangent linear and adjoint model. This 4DEnVar method has been used to estimate JULES LSM crop parameters at a single Nebraskan site Pinnington et al. (2020) and to calibrate pedotransfer functions to improve JULES LSM soil moisture predictions over East Anglia (Pinnington et al., 2021) and the whole of the UK (Cooper

et al., 2021).

The problem addressed in this article is the assimilation of atmospheric $CO_2$ data to calibrate the parameters of the OR-CHIDEE LSM. For this application, we need to couple ORCHIDEE with an atmospheric transport model, which, in our case, is LMDZ, as they are historically linked and represent the land and atmospheric components of the IPSL (Institut Pierre-

Simon-Laplace) Earth system model (Boucher et al., 2020). While tangent linear and adjoint models can be easily derived for the transport model (Hourdin et al., 2006; Hourdin and Talagrand, 2006), this is not the case for the ORCHIDEE LSM.





Although tangent linear or adjoint models are not required for methods such as GA, MCMC, or emulator-based approaches , they necessitate defining a large ensemble, they are unfeasible for use in this study due to the time-consuming nature of model simulations. The purpose of this article is to present an adjoint-free data assimilation framework that facilitates the assimilation

of atmospheric $CO_2$ concentrations. We demonstrate the potential of 4DEnVar using synthetic observational data and compare its performance with that of 4DVar with finite differences. Section 2 presents the methods, the models, the data and the experiments. Results are shown in Section 3, with discussions and conclusions in Sections 4 and 5, respectively.

## 2   Method

### 2.1   Models and datasets

#### 2.1.1   ORCHIDEE land surface model

ORCHIDEE (ORganizing Carbon and Hydrology In Dynamic EcosystEms; originally described in Krinner et al. (2005)) is a process-based LSM that simulates the exchange of carbon, water and energy between the surface, vegetation, and the atmosphere. It is composed of different sub-models: a fast one that calculates photosynthesis, hydrology and energy balance every 30 minutes; and a slow one that simulates carbon allocation in plant reservoirs, soil carbon dynamics and litter decomposi-

tion every day. In this study, we used the ORCHIDEE version 2 used in the Coupled Model Intercomparison Project Phase 6 (CMIP6) (Boucher et al., 2020; Lurton et al., 2020). This version contains significant improvements over the original version described by Krinner et al. (2005). The soil hydrology scheme is based on Richards' equation that describes vertical water fluxes for a soil depth of 2 m discretised into 11 layers (de Rosnay et al., 2002). The vertical discretisation for heat diffusion is identical to that used for water up to 2 m extended to 90 m with a zero flux condition at the bottom and with 18 calculation

nodes in order to extrapolate the water content across the entire profile between 2 m and 90 m (Wang et al., 2016). The hydrological and thermal properties of the soil are determined by soil moisture and texture. The dominant soil texture for each model grid cell is derived from the ZOBLER map (Zobler, 1999) using a classification system comprising 3 categories. The set of equations governing the Soil Organic Matter (SOM) pools and their temporal evolution have analytical solution driven by litter input and climate conditions, including soil temperature and humidity (Lardy et al., 2011).

The carbon assimilation scheme follows the approach presented by Yin and Struik (2009) based on the FvCB model (Farquhar et al., 1980) for C3 plants and Collatz et al. (1991) for C4 plants. The ORCHIDEE LSM uses different types of vegetation grouped into Plant Functional Types (PFT) with similar structural characteristics. It distinguishes 14 vegetation PFT classes described in Table A1. Each grid point in the model is associated with PFT fractions prescribed using annually varying PFT maps derived from ESA's Climate Change Initiative land cover (LC) products and a LC-to-PFT cross-walking approach (Poulter

et al., 2015) (see https://orchidas.lsce.ipsl.fr/dev/lccci/).

In this study, ORCHIDEE is run offline using 3-hour ERA-Interim surface weather forcing fields (Dee et al., 2011) over 2000-2001, and aggregated to the spatial resolution of the LMDZ atmospheric transport model (2.5° latitude × 3.75° longitude) . The carbon pools are brought to equilibrium following the TRENDY protocol (Sitch et al., 2024). This involves spinning up





the model for 200 years, employing an analytical spin-up for soil carbon pools to bring them to equilibrium. This process uses a
constant $CO_2$ concentration of 1700, no land-use change (LUC), and recycled ERA-Interim meteorological data from 1990 to
1999, as these are the only years where forcing data is available preceding the assimilation period. This spin-up run is followed
by a transient simulation to account for the effects of disturbances, varying global atmospheric $CO_2$ concentration and LUC
from 1800 to 1999, recycling the same meteorological data.

### 2.1.2 LMDZ atmospheric transport model

The atmospheric transport model used in this study is version 3 of the LMDZ General Circulation Model (GCM) (Hourdin
and Armengaud, 1999). The LMDZ atmospheric model has been widely used to model the climate; it was implemented as
the atmospheric component of the IPSL Earth System Model (Dufresne et al., 2013). Its derived transport model has been
used to simulate gas, particle chemistry and greenhouse gas distributions in numerous studies (Peylin et al., 2005; Chevallier
et al., 2005; Locatelli et al., 2015; Remaud et al., 2018). The advection is based on Van Leer scheme (Leer, 1977); the deep
convection is parametrised following the scheme of (Tiedtke, 1989); and turbulent mixing in the planetary boundary layer
is based on a second-order local closure formalism (Hourdin and Armengaud, 1999). It uses a horizontal resolution of 2.5°
(latitude) × 3.75° (longitude) and 19 sigma-pressure layers up to 3 hPa. In this study, we use pre-calculated transport fields,
as described in Peylin et al. (2005): they quantify the sensitivity of atmospheric concentrations at a given atmospheric station
according to the space-time variability of the surface fluxes. The temporal resolution of the concentration is monthly, taking
into account the daily surface fluxes of each grid cell in the model (as shown on the Fig. 1). These pre-calculated transport
fields have proven to be very useful - they considerably reduce computing time, given that the model only needs to be run
once. They have been used to assimilate atmospheric $CO_2$ data in a few data assimilation studies (Peylin et al., 2016; Bacour
et al., 2023). Although the version of LMDZ used in this study is outdated, the main objective of this work is to develop a
framework for atmospheric data assimilation that will support future research using an updated version of LMDZ. Therefore,
these pre-calculated transport fields provide a low-cost experiment so as to address the methodological and technical challenges
that were previously presented. The pre-calculated transport fields were originally calculated to assimilate atmospheric $CO_2$
concentration data using the NOAA Earth System Laboratory's collaborative product (GLOBALVIEW-CO2, 2013). They
model average monthly concentrations at 53 stations over the period 1990-2009. The stations are located at different altitudes
and in different locations on the continents and oceans around the world.

### 2.1.3 Atmospheric stations

Fig. 1 shows the location of the 21 stations selected for this study. The stations were selected according to their sensitivity to
continental fluxes (also shown in Fig. 1) in order to capture the temporal and spatial variations in fluxes over the continental
surface. The selected stations are therefore mainly located above the land surface. The other stations, mainly located over the
oceans, are less sensitive to continental fluxes, capturing mainly long term variations. As we are only assimilating 2 years of
concentrations, we choose not to take them into account.





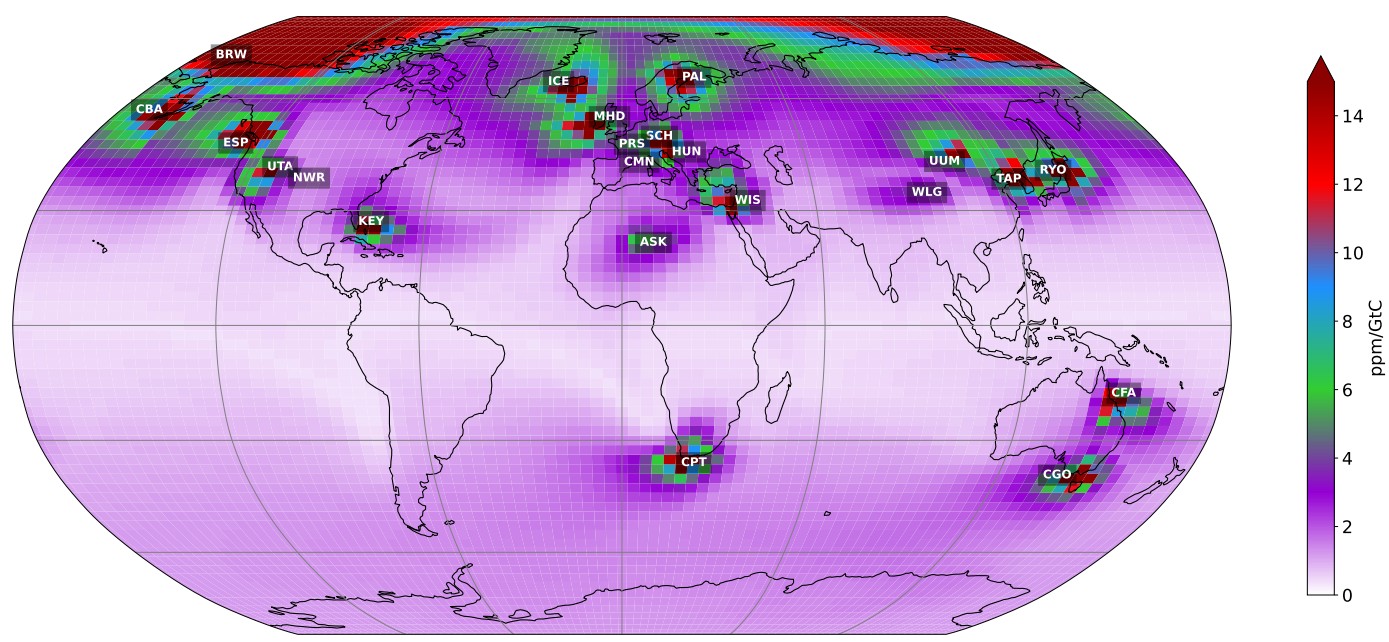

**Figure 1.** 6-month averaged sensitivity of atmospheric $CO_2$ concentration in ppm/GtC at each atmospheric station according to surface fluxes. The colour of the pixel indicates the influence of the surface fluxes given by the pixel on the atmospheric concentration of $CO_2$, depending on the station. Red indicates a very strong influence of surface fluxes. The blue, green and violet colours indicate different influences, from strong to weak. White indicates no influence from surface fluxes (see full detail of the stations https://gml.noaa.gov/dv/site/index.php).

### 2.1.4 Other components of surface $CO_2$ fluxes

Other components contributing to the global surface fluxes are not optimised in this study:

- The oceanic flux component was derived from a neural networks model which estimated the spatial and temporal variations in $CO_2$ fluxes between the air and the sea (Peylin et al., 2016).

- The global maps of biomass burning emissions are taken from the Global Fire Emission Database version 3 (Randerson et al., 2013).





– The fossil fuel $CO_2$ emission products used here were developed by the University of Stuttgart/IER on the basis of EDGAR v4.2.

All the fluxes used are described in greater detail in previous studies (Peylin et al., 2016; Bacour et al., 2023) and are shown in Fig. A1.

## 2.2 Data assimilation framework

### 2.2.1 A Bayesian setup

First, let us define a general Bayesian framework, mainly following Tarantola (1987, 2005), that accounts both for model/observation error and an *a priori* background error. Taking the approach of Kennedy and O'Hagan (2001), for an observational constraint $\mathbf{y}$, let

$$\mathbf{y} = \mathcal{Y} + \mathbf{e} \tag{1}$$

where $\mathcal{Y}$ represents the relevant aspect of the observed system and $\mathbf{e}$ represents the error on that observation, often due to instrument error but can include any error in the derivation of the data product. Let $\mathcal{H}$ represent the model operator that takes the parameter vector $\mathbf{x}$ as input. We then assume that there exists an input $\mathbf{x}^*$ such that:

$$\mathbf{y} = \mathcal{Y} + \mathbf{e} = \mathcal{H}(\mathbf{x}^*) + \boldsymbol{\eta} + \mathbf{e} \tag{2}$$

where $\boldsymbol{\eta}$ represents the model error, given an imperfect model. Note that, given no additional information about the errors, we assume that i) $\mathbf{e}$ and $\boldsymbol{\eta}$ are independent of $\mathcal{Y}$ and $\mathcal{H}(\mathbf{x})$ respectively and ii) both are random vector quantities following a multivariate normal distribution with a mean equal to 0 and a covariance matrix $\boldsymbol{\Sigma}_i$ such that $\mathbf{e} \sim \mathcal{N}(0, \boldsymbol{\Sigma_e})$ and $\boldsymbol{\eta} \sim \mathcal{N}(0, \boldsymbol{\Sigma_\eta})$. Furthermore, we assume that the parameter vector $\mathbf{x}$ and the model/observation likelihood $\mathbf{y}|\mathbf{x}$ both follow Gaussian multivariate distributions:

$$p(\mathbf{y}|\mathbf{x}) \propto \exp\left[-\frac{1}{2}(\mathcal{H}(\mathbf{x}) - \mathbf{y})^T \mathbf{R}^{-1}(\mathcal{H}(\mathbf{x}) - \mathbf{y})\right]; \qquad p(\mathbf{x}) \propto \exp\left[-\frac{1}{2}(\mathbf{x} - \mathbf{x_b})^T \mathbf{B}^{-1}(\mathbf{x} - \mathbf{x_b})\right], \tag{3}$$

where $\mathbf{x_b}$ represents prior knowledge of the parameter vector and $\mathbf{B}$ and $\mathbf{R}$ are respectively the covariance error matrix for the parameters vector and for the model/observation such that $\mathbf{R} = \boldsymbol{\Sigma_\eta} + \boldsymbol{\Sigma_e}$. We seek to find the posterior distribution $p(\mathbf{x}|\mathbf{y})$ which quantifies the probability of parameters given the observations using Bayes' theorem:

$$p(\mathbf{x}|\mathbf{y}) \propto p(\mathbf{y}|\mathbf{x})p(\mathbf{x}) \propto \exp\left[-\frac{1}{2}(\mathcal{H}(\mathbf{x}) - \mathbf{y})^T \mathbf{R}^{-1}(\mathcal{H}(\mathbf{x}) - \mathbf{y}) - \frac{1}{2}(\mathbf{x} - \mathbf{x_b})^T \mathbf{B}^{-1}(\mathbf{x} - \mathbf{x_b})\right]. \tag{4}$$

### 2.2.2 The 4DVar method

**Standard 4DVar**

In this Section, we present the 4DVar assimilation method. Maximising the probability in equation (4) is equivalent to minimising the following function, usually referred to as the 4DVar *cost function*:

$$J(\mathbf{x}) = \frac{1}{2}(\mathcal{H}(\mathbf{x}) - \mathbf{y})^T \mathbf{R}^{-1}(\mathcal{H}(\mathbf{x}) - \mathbf{y}) + \frac{1}{2}(\mathbf{x} - \mathbf{x_b})^T \mathbf{B}^{-1}(\mathbf{x} - \mathbf{x_b}). \tag{5}$$





In 4DVar minimisation, the observation vector $\mathbf{y}$ is assimilated over the entire time window (compared to 3DVar where observations are compared to a single-model output). The minimum can be reached iteratively using a descent algorithm that requires the computation of the gradient of $J$ with respect to the parameter vector $\mathbf{x}$. In addition, when the model is non-linear, it is common to use the quasi-Newton method to optimise the parameters vector:

$$\mathbf{x}_{i+1} = \mathbf{x}_i - (\nabla^2 J(\mathbf{x}_i))^{-1} \times \nabla J(\mathbf{x}_i). \tag{6}$$

The gradient of the cost function, $\nabla J(\mathbf{x}_i)$, and the square matrix of partial second derivatives of the cost function (called the Hessian matrix), $\nabla^2 J(\mathbf{x}_i)$, can be calculated as follows:

$$\nabla J(\mathbf{x}_i) = \mathbf{H}^T \mathbf{R}^{-1}(\mathbf{H}\mathbf{x}_i - \mathbf{y}) + \mathbf{B}^{-1}(\mathbf{x}_i - \mathbf{x_b}); \qquad \nabla^2 J(\mathbf{x}_i) = \mathbf{H}^T \mathbf{R}^{-1}\mathbf{H} + \mathbf{B}^{-1}. \tag{7}$$

We can update equation (6) using (7):

$$\mathbf{x}_{i+1} = \mathbf{x}_i - [\mathbf{H}^T \mathbf{R}^{-1}\mathbf{H} + \mathbf{B}^{-1}]^{-1}[\mathbf{H}^T \mathbf{R}^{-1}(\mathbf{H}\mathbf{x}_i - \mathbf{y}) + \mathbf{B}^{-1}(\mathbf{x}_i - \mathbf{x_b})]. \tag{8}$$

Here, the notation $\mathcal{H}$ becomes $\mathbf{H}$, because it does not represent the use of the direct operator $\mathcal{H}$. Instead, we use the tangent linear model $\mathbf{H}$ and the adjoint model $\mathbf{H}^T$. Usually, these two terms are coded directly, but for complex models, it is usually very difficult to code and maintain these terms, especially when the model is subject to many developments (which means that they quickly become obsolete).

**Epsilon-based 4DVar variant: $\epsilon$-4DVar**

To approximate the tangent linear and adjoint models, we can use finite differences:

$$\mathbf{H} = \frac{\mathcal{H}(\mathbf{x} + \Delta\mathbf{x}) - \mathcal{H}(\mathbf{x})}{\Delta\mathbf{x}} \tag{9}$$

where $\Delta\mathbf{x}$ represents a small change in $\mathbf{x}$. This estimate will not be as accurate as the exact tangent linear and adjoint models, but it can still help us in our minimisation objective. The accuracy of the tangent linear and adjoint models is then completely dependent on the choice of $\Delta\mathbf{x}$. A selection of $\Delta\mathbf{x}$ that is too small may lead to $\mathbf{H}$ being insensitive to the parameter vector, i.e. $\mathcal{H}(\mathbf{x} + \Delta\mathbf{x}) - \mathcal{H}(\mathbf{x}) = \Delta\mathbf{x}\mathbf{H} \approx \mathbf{0}$. This leads to the term corresponding to the difference between the observation and the output's operator $(\mathbf{H}\mathbf{x}_i - \mathbf{y})$ becoming negligible in equation (7) and hence resulting in an ineffective minimisation. By contrast, if the choice of $\Delta\mathbf{x}$ is too large, the result gives inaccurate tangent linear and adjoint models that lose their local vision around $\mathbf{x}$. This results in a large loss of information and therefore a much less accurate minimisation. In our case, we define $\epsilon$ such that: $\Delta\mathbf{x} = \mathbf{x_{range}} * \epsilon$, where $\mathbf{x_{range}} = \mathbf{x_{max}} - \mathbf{x_{min}}$ and we will refer to this method as *$\epsilon$-4DVar*.

### 2.2.3  The 4DEnVar method

**From 4DVar to 4DEnVar**

We present here an implementation of 4DVar that we do not use in this study, but that is important to understand the 4DEnVar method. This implementation is presented in several studies (Courtier et al., 1994; Gilbert and Lemaréchal, 1989; Liu et al., 2008; Bannister, 2017; Pinnington et al., 2020), and can be applied when the prior error covariance matrix $\mathbf{B}$ becomes large



and difficult to invert. It is possible to introduce a matrix $\mathbf{U}$ and a vector $\mathbf{w}$ to ensure that the 4DVar cost function converges as efficiently as possible and avoids the explicit calculation of the matrix $\mathbf{B}$ given by:

$$\mathbf{B} = \mathbf{U}\mathbf{U}^T \tag{10}$$

and

$$\mathbf{x_a} = \mathbf{x_b} + \mathbf{U}\mathbf{w} \tag{11}$$

where $\mathbf{x_a}$ represents the posterior value of the parameter vector. Consequently, this changes the $J$ cost function, which is presented in detail in Courtier et al. (1994):

$$J(\mathbf{w}) = \frac{1}{2}(\mathbf{HUw} + \mathcal{H}(\mathbf{x_b}) - \mathbf{y})^T \mathbf{R}^{-1}(\mathbf{HUw} + \mathcal{H}(\mathbf{x_b}) - \mathbf{y}) + \frac{1}{2}\mathbf{w}^T\mathbf{w} \tag{12}$$

and its gradient:

$$\nabla J(\mathbf{w}) = \mathbf{U}^T\mathbf{H}^T\mathbf{R}^{-1}(\mathbf{HUw} + \mathcal{H}(\mathbf{x_b}) - \mathbf{y}) + \mathbf{w}. \tag{13}$$

**4DEnVar**

The 4DEnVar method described in Liu et al. (2008) and Pinnington et al. (2020) proposes to incorporate an aspect of the ensemble Kalman Filter (EnKF) in order to avoid the calculation of tangent linear or adjoint models necessary for 4DVar. The EnKF is a Kalman filter, but uses a set of $N$ parameter vectors, also known as ensemble members, to estimate the prior error covariance matrix $\mathbf{B}$ (Evensen, 1994). A perturbation matrix:

$$\mathbf{X'_b} = \frac{1}{\sqrt{N-1}}\left(\mathbf{x_1} - \mathbf{x_b}; \mathbf{x_2} - \mathbf{x_b}; ...; \mathbf{x_N} - \mathbf{x_b}\right). \tag{14}$$

where the ensemble members $\mathbf{x}_i$ for $i = 1, ..., N$ are generated according to a multivariate normal distribution using $\mathbf{x_b}$ as the mean and $\mathbf{B}$ as the covariance matrix: $\mathcal{N}(\mathbf{x_b}, \mathbf{B})$. It follows that:

$$\mathbf{B} \approx \mathbf{X'_B}\mathbf{X'_B}^T. \tag{15}$$

Using the same logic as equation (11), we can use the perturbation matrix as follows:

$$\mathbf{x_a} = \mathbf{x_b} + \mathbf{X'_b}\mathbf{w} \tag{16}$$

where $\mathbf{w}$ is a vector of length $N$. The cost function in Equation (12) is updated accordingly:

$$J(\mathbf{w}) = \frac{1}{2}(\mathbf{HX'_b}\mathbf{w} + \mathcal{H}(\mathbf{x_b}) - \mathbf{y})^T \mathbf{R}^{-1}(\mathbf{HX'_b}\mathbf{w} + \mathcal{H}(\mathbf{x_b}) - \mathbf{y}) + \frac{1}{2}\mathbf{w}^T\mathbf{w}, \tag{17}$$

and the gradient in Equation (13) becomes:

$$\nabla J(\mathbf{w}) = \mathbf{X'_b}^T\mathbf{H}^T\mathbf{R}^{-1}(\mathbf{HX'_b}\mathbf{w} + \mathcal{H}(\mathbf{x_b}) - \mathbf{y}) + \mathbf{w}. \tag{18}$$





Note that the minimisation problem changes. In both case we try to balance the cost function between the background term and the observation term but we no longer aim to find $\mathbf{x}$ such that $\mathcal{H}(\mathbf{x}) \approx \mathbf{y}$, but we now look for $\mathbf{w}$ that determines the linear combination $\mathbf{HX}'_\mathbf{b}\mathbf{w}$ which is equal to the distance $\delta\mathbf{y}$ such that $\delta\mathbf{y} \approx \mathcal{H}(\mathbf{x}) - \mathbf{y}$. The $\mathbf{HX}'_\mathbf{b}$ term can be approximated by applying the $\mathcal{H}$ operator to each parameter vector $\mathbf{x}$ present in $\mathbf{X}'_\mathbf{b}$ :

$$\mathbf{HX}'_\mathbf{b} \approx \frac{1}{\sqrt{N-1}} \left( \mathcal{H}(\mathbf{x_1}) - \mathcal{H}(\mathbf{x_b}); \mathcal{H}(\mathbf{x_2}) - \mathcal{H}(\mathbf{x_b}); ...; \mathcal{H}(\mathbf{x_N}) - \mathcal{H}(\mathbf{x_b}) \right). \tag{19}$$

Each coefficient $w_i$ of $\mathbf{w}$ multiplies a vector $\mathcal{H}(\mathbf{x}_i) - \mathcal{H}(\mathbf{x_b})$ present in the approximation of $\mathbf{HX}'_\mathbf{b}$ which represents the distance between a member of the ensemble and the prior information. The optimisation of $\mathbf{w}$ is performed so that the linear combination $\mathbf{HX}'_\mathbf{b}\mathbf{w}$ converges around $\delta\mathbf{y}$ and taking into account the background terms. Once optimised, the vector $\mathbf{w}$ can be used for another linear combination $\mathbf{X}'_\mathbf{b}\mathbf{w}$, this time in the input space. This gives $\mathbf{x_a}$, the posterior value of the parameter vector, that can be obtained using equation (16). The great advantage of this method lies in the way the gradient is computed. In particular, the term $\mathbf{X}'_\mathbf{b}{}^T\mathbf{H}^T$, which is equivalent to $(\mathbf{HX}'_\mathbf{b})^T$. This equivalence makes it possible to rewrite the gradient by "simply" transposing the matrix $\mathbf{HX}'_\mathbf{b}$ :

$$\nabla J(\mathbf{w}) = (\mathbf{HX}'_\mathbf{b})^T \mathbf{R}^{-1} (\mathbf{HX}'_\mathbf{b}\mathbf{w} + \mathcal{H}(\mathbf{x_b}) - \mathbf{y}) + \mathbf{w}. \tag{20}$$

Subsequently, tangent linear and adjoint models are no longer required. The subjective choice here is no longer related to the choice of the $\epsilon$ that estimates the tangent linear and adjoint models, but to the number $N$ of ensemble members used to generate $\mathbf{X}'_\mathbf{b}$ and $\mathbf{HX}'_\mathbf{b}$.

### 2.2.4 Implementation into ORCHIDAS

The ORCHIDEE Data Assimilation System (ORCHIDAS) is a system desgined to calibrate the parameters of ORCHIDEE and is developed in Python. It has been used for over 15 years (MacBean et al., 2022) mainly for studies focusing on the carbon cycle and other terrestrial cycles such as water and energy budget, methane and nitrogen (see the full list of studies published at https://orchidas.lsce.ipsl.fr/publications.php).

This system has long used 4DVar as described in Section 2.2.2, but it also allows the use of several methods such as genetic algorithms (Bastrikov et al., 2018) or history matching (Raoult et al., 2024a). ORCHIDAS facilitates the testing of various data assimilation methods while maintaining a consistent configuration for ORCHIDEE execution. In this study, we have implemented the 4DEnVar method as described in Section 2.2.3.

### 2.3 Twin experiments

To test the two methods presented in Section 2.2, we use a so-called *twin experiment*. This experiment eliminates all the complexities associated with model-data errors by having $\mathbf{e} \approx \boldsymbol{\eta} \approx 0$ in Equation 2 and focuses on the efficiency of the assimilation method. In this *twin experiment*, we aim to optimise the net biome productivity (NBP) fluxes of the ORCHIDEE LSM by calibrating the parameters involved in their calculation. The NBP fluxes represent the net carbon fluxes of the land component, i.e. the difference between the emission fluxes of heterotrophic and autotrophic respiration as well as the disturbance fluxes





due to LUC and the sink fluxes mainly due to photosynthesis. To do this, we simulate the NBP fluxes at the global scale using the ORCHIDEE LSM with the default parameter values - which we will refer to as 'true' in the remainder of this article - and include the other fluxes described in Section 2.1.4 over the year 2000-2001. We then transport the concentration given

by the surface fluxes using the pre-calculated transport fields of LMDZ. We focus on 21 continental atmospheric stations, as shown in Fig. 1. These stations are highly sensitive to carbon fluxes over the continents, which provides a significant constraint on continental fluxes and therefore on our parameters. This has enabled us to generate 'synthetic' observations of monthly average atmospheric $CO_2$ concentrations given by these 21 atmospheric stations between 2000 and 2001. A limited period was chosen for practical reasons - to avoid expensive simulations. A new *a priori* parameter vector was generated, different

from the 'true' parameter values. We then applied the various assimilation methods to see how closely they converge towards the known solution (standard parameter values). The assimilation of the atmospheric $CO_2$ concentration at the 21 stations is carried out simultaneously.

### 2.3.1   Simplified case

First, we focus on a simplified case involving the calibration of only one PFT-dependant parameter: $Vcmax$, which controls

the maximum rate of carboxylation limited by Rubisco activity at $25°C$. This parameter was chosen because its impact on the atmospheric $CO_2$ concentration is well understood: when its value increases, the quantity of carbon absorbed by photosynthesis increases and atmospheric concentrations decrease - and vice versa. The aim of the assimilation is to recover the 'true' values of $Vcmax$ for the 14 PFTs resulting in the calibration of 14 parameters. This simplified case is very useful to perform several tests allowing for a better understanding of the behaviour of the different data assimilation methods.

### 2.3.2   Complex case

To assess the performance of the different approaches in conditions resembling real cases, we perform another twin experiment in which we calibrate four PFT-dependent parameters and one global parameter involved in different bio-geophysical processes. The parameters selected have already been optimised in previous data assimilation studies using atmospheric $CO_2$ concentrations (Peylin et al., 2016; Bacour et al., 2023). In addition to $Vcmax$, we choose:

– the PFT-dependent parameter **SLA** (Specific Leaf Area) that impacts leaf biomass and hence ecosystem photosynthetic capacity;

– the global parameter $Q10$ which controls the thermal dependence of heterotrophic respiration;

– the PFT-dependent parameter $m_{maint.resp}$ that defines the slope of the maintenance respiration coefficient, which controls autotrophic respiration;

– the PFT-dependent parameter $LAI_{max}$ which controls the maximum leaf area index for carbon allocation. It impacts the vegetation biomass and therefore acts on both photosynthesis and respiration.





A total of 57 $(14 \times 4 + 1)$ parameters are being calibrated. As they interact within the same modelled processes, the degree of equifinality is significant.

### 2.3.3 Error covariance matrices

We need to define the two error covariance matrices, $\mathbf{R}$ and $\mathbf{B}$, in order to use the two data assimilation methods. Since we are assimilating 'synthetic' observations, these matrices can be diagonal. The $\mathbf{R}$ matrix is used to represent the model/observation error. In our case, we define small diagonal terms of $0.01$ ppm for the $\mathbf{R}$ matrix as we are in a perfect model scenario. The $\mathbf{B}$ matrix contains the background errors associated with the prior knowledge of the parameters. We set an error corresponding to $30\%$ of the parameter range for the simple case and $20\%$ for the complex case (as we use larger parameter ranges). To ensure
that the experiments are comparable, the $\mathbf{R}$ and $\mathbf{B}$ matrices are common to the two methods: $\epsilon$-4DVar and 4DEnVar .

### 2.4 Tuning $\epsilon$ for gradient calculation

As explained in Section 2.2.2, the choice of $\epsilon$ is essential for effective $\epsilon$-4DVar performance. One way to select an appropriate $\epsilon$ is to perform a $\epsilon$-test which calculates the partial derivative of $\mathcal{H}$ for each of the parameters and using different $\epsilon$. We calculate the partial derivative as follows:

$$\frac{\partial \mathcal{H}}{\partial \mathbf{x}} = \frac{\mathcal{H}(\mathbf{x} + \Delta \mathbf{x}) - \mathcal{H}(\mathbf{x})}{\Delta \mathbf{x}} \quad (21)$$

where $\epsilon$ defines $\Delta \mathbf{x}$ as explained in Section 2.2.2. By changing the $\epsilon$ we change $\Delta \mathbf{x}$ and we can seek to find the value of $\epsilon$ for which the derivative becomes stable. Fig. 2 shows the sensitivity of $\epsilon$ ranging from $10^{-8}$ to $10^{-2}$ on the calculates the partial derivative of each $\boldsymbol{Vcmax}$. We see that the partial derivative of $\boldsymbol{Vcmax}$ is unstable with an $\epsilon$ below $10^{-3}$ for all PFT. Therefore, we need a value of $\epsilon$ greater than $10^{-3}$ to ensure correct gradient calculation with respect to the $\boldsymbol{Vcmax}$
parameter. Table A2 shows the values of the mean of the partial derivatives for all parameters and PFTs using an $\epsilon$ allowing for a stable derivative. This also allows us to check the consistency of the derivation calculation. For example, the increase in $\boldsymbol{Vcmax}$ leads to an increase in the photosynthetic capacity and subsequently in the carbon uptake by vegetations. This leads to a reduction in atmospheric $CO_2$ concentration. We can see in Table A2 that the values obtained for $\boldsymbol{Vcmax}$ are negative which is the expected response. The same $\epsilon$-test was carried out for the four other parameters used in the complex case, and
the results are shown in Fig. A2 and in Table A2 :

- The partial derivative of $\mathbf{SLA}$ diverges with an $\epsilon$ below $10^{-3}$ for all PFT. $\mathbf{SLA}$ has the same impact that $\boldsymbol{Vcmax}$ has on atmospheric $C0_2$ concentration, so the negative mean values obtained are expected;

- The partial derivative of $\mathbf{Q10}$ does not diverge for any values of $\epsilon$. The mean value of its derivation is negative as expected. Increasing $\mathbf{Q10}$ increases the thermal dependence of heterotrophic respiration and consequently reduces it;
with less heterotrophic respiration the atmospheric $CO_2$ concentration decreases;

- The partial derivative of $\boldsymbol{m_{maint.resp}}$ diverges with different $\epsilon$ values depending on the PFT, ranging from $10^{-5}$ for PFT TrBE to $10^{-2}$ for PFT CropsC$_4$. This may be due to different distributions and proportions of PFTs (see Table A1).





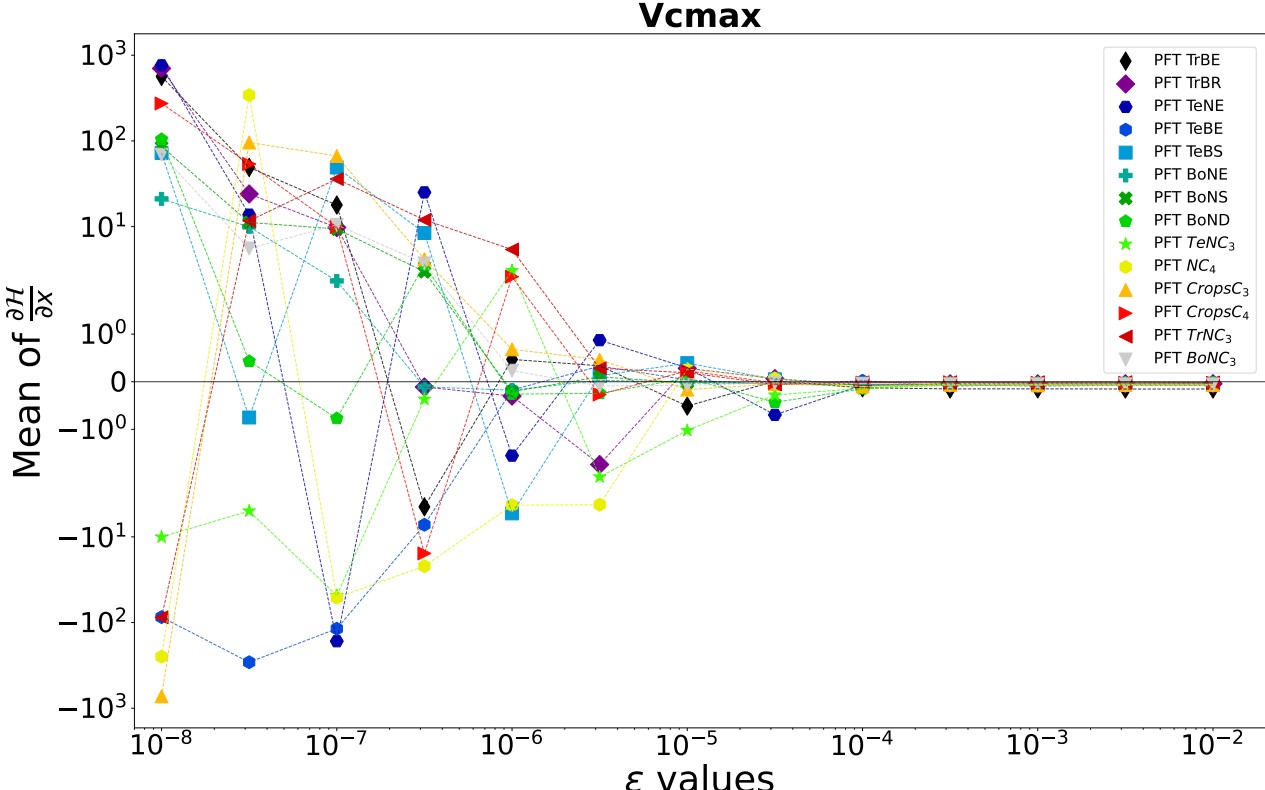

**Figure 2.** $\epsilon$-test: Mean of partial derivative of $\mathcal{H}$ as a function of $\epsilon$. The partial derivative of $\mathcal{H}$ is calculated with respect to the parameter $\boldsymbol{Vcmax}$ for each PFT. It is calculated on the concentration space using every station over 2 years. The mean of the partial derivative is then calculated over space and time in order to visualise the local derivative. The derivative of $\mathcal{H}$ is calculated for several $\epsilon$.

However, the mean values at $10^{-2}$ are all positive. $m_{maint.resp}$ has an impact on autotrophic respiration, increasing this parameter increases vegetation respiration and therefore the atmospheric $CO_2$ concentration;

– The partial derivative of $LAI_{max}$ diverges with an $\epsilon$ below $10^{-2}$ for all PFT. Determining the sign of the mean values of the partial derivative of this parameter is not trivial here. $LAI_{max}$ influences vegetation biomass and therefore photosynthesis and respiration. All PFTs gives a negative mean values for their partial derivative, only the PFT TrBR gives a positive mean value.

**2.5 Defining the impact of the configuration**

For both methods, $\epsilon$-4DVar and 4DEnVar, the configuration used plays an important role in the quality of the minimisation of the associated cost function and so the calibration of the parameter. Whether it is the choice of $\epsilon$ for the $\epsilon$-4DVar or the number





of members used to generate the ensemble in the 4DEnVar, it is up to the user to make a choice that can only be subjective. To assess their impact, we launch the twin experiment using different configurations:

- for the simple case:

    - 5 different values of $\epsilon$ for the $\epsilon$-4DVar based on the sensitivity test presented in Section 2.4 ;

    - 5 different ensemble sizes in the 4DEnVar.

- for the complex case:

    - 5 different ensemble sizes in the 4DEnVar.

For the complex case using $\epsilon$-4DVar , the $\epsilon$ is selected relative to the results in the simple case and Fig. A2. To re-tune $\epsilon$ for each parameter requires too many simulations and so it is not feasible for the complex case.

For each minimisation, the L-BFGS-B (limited memory Broyden–Fletcher–Goldfarb– Shanno) algorithm with bound constraints; (Byrd et al., 1995)) algorithm is used. For the $\epsilon$-4DVar, we set a maximum number of iterations at $40$, due to computing costs. Indeed, each iteration requires $N_{param}+1$ model simulations. In each case, a solution is reached after $20$ iterations (subsequent iterations are only minor corrections of the solution obtained). For 4DEnVar, no maximum iteration limit is chosen, since an iteration does not requires further simulation of the model (all required information is contained in the pre-calculated ensemble). We can therefore wait for the L-BFGS-B minimiser to converge, i.e. until the gradient becomes null.

## 3 Results

### 3.1 Comparing the different configurations

The results in terms of 1) mean reduction in root mean square difference (RMSD) of the 21 atmospheric stations, 2) Mean Absolute Differences (MAD) on parameter space and 3) computational demand of each experiment using the simple case are summarised in Table 1. We see that for the $\epsilon$-4DVar method the best results are obtained with an $\epsilon$ equal to $5*10^{-2}$ where the mean RMSD reduction is $82.3\%$ and the MAD score is $1.7$. The best results for the 4DEnVar method are obtained using an ensemble of $100$ members where the mean RMSD reduction is $97\%$ and the MAD score is $0.3$. These two configurations are therefore considered for the simple case of the twin experiment in Section 3.2.1.

For the complex case, results are presented in Table 2. We see that the best results for the 4DEnVar method are obtained with an ensemble of $300$ members giving a mean RMSD reduction of $94.4\%$. This configuration is considered for the complex case in Section 3.2.2 using the 4DEnVar method.





**Table 1.** Mean RMSD reduction score between "synthetic" observation and posterior simulation of the atmospheric $CO_2$ concentration at 21 atmospheric stations, Mean Absolute Difference (MAD) score computed between the "true" parameter values used to generate the "synthetic" observations and the posterior parameters and number of simulations used for each configuration of $\epsilon$-4DVar and 4DEnVar for the simple case.

| $\epsilon$-4DVar | *Epsilon* | Mean RMSD reduction | MAD score | Number of ORCHIDEE simulations needed |
|---|---|---|---|---|
| | $10^{-1}$ | 79.7% | 1.84 | 300 |
| | $5*10^{-2}$ | 82.3% | 1.7 | 300 |
| | $10^{-2}$ | 75.1% | 2.05 | 300 |
| | $5*10^{-3}$ | 73.1% | 1.91 | 300 |
| | $10^{-3}$ | 69.5% | 2.0 | 300 |
| **4DEnVar** | *Ensemble* | | | |
| | 50 | 81.1% | 0.44 | 50 |
| | 75 | 91.8% | 0.67 | 75 |
| | 100 | 97.0% | 0.3 | 100 |
| | 150 | 91.0% | 0.34 | 150 |
| | 200 | 96.3% | 0.29 | 200 |

**Table 2.** Mean RMSD reduction score between "synthetic" observations and posterior simulation of the atmospheric $CO_2$ concentration at 21 atmospheric stations for each configuration of the 4DEnVar method for the complex case

| 4DEnVar | *Ensemble* | Mean RMSD reduction |
|---|---|---|
| | 100 | 81% |
| | 200 | 89.8% |
| | 300 | 94.4% |
| | 350 | 94.0% |
| | 400 | 90.9% |

## 3.2 Comparing $\epsilon$-4DVar and 4DEnVar

### 3.2.1 Simple case

375  Fig. 3 and Fig. 4 compare the results obtained for the $\epsilon$-4DVar and the 4DEnVar methods using the configurations chosen in Section 3.1. Fig. 3 shows that the parameter values obtained by the 4DEnVar method is almost equal to the "true" parameters used to generate the "synthetic" observations with a mean absolute difference (MAD) score of 0.3. This shows that the 4DEnVar method is able to almost recover the "true" parameters. The parameter values obtained by the $\epsilon$-4DVar method have a MAD score of 2.05, which reduces the prior MAD by 30% but remains far from the "true" parameter values. Only the $Vcmax$

380  of PFTs TeNC$_3$, TeNE and TrBE are close to the "true" value of the parameters whereas PFTs BoNC$_3$, BoNE, TeBS, TeBE





and TrBR give values that are between the prior and the "true" value; the $\boldsymbol{Vcmax}$ of other PFTs have either maintained or increased the distance between the prior and the "true" values. This shows that the $\epsilon$-4DVar method falls into a local minimum and is therefore unable to recover the "true" parameters. Fig. 4 shows the different RMSD scores between the synthetic observations and prior/posterior simulations for each of the 21 atmospheric stations. The average reduction of RMSD for $\epsilon$-4DVar methods is $82\%$ with a mean RMSD of $0.1$ ppm. The largest reduction of RMSD is for the German station, Shauinsland (SCH) ($87\%$) and the lowest is for the Australian station, Cape Grim (CGO) ($49.8\%$). Comparatively, the average reduction of RMSD for 4DEnVar methods reaches $97\%$ with a mean RMSD of $0.01$ ppm across all stations. The highest reduction of RMSD is for the Chinese station, Waliguan (WLG) ($99\%$) and the lowest one is for the Australian station, Cape Cleveland (CFA) ($92.7\%$). We see that the 4DEnVar method outperforms the $\epsilon$-4DVar method: the 4DEnVar method has the best fit to the "synthetic" observations assimilated and can find the value of the "true" parameters used to generate the "synthetic" observations.

### 3.2.2 Complex case

For the complex case, Fig. 5 shows the prior/posterior RMSD at each atmospheric station for the 4DEnVAR method using a ensemble of 300 members and the $\epsilon$-4DVar method using an $\epsilon$ of $5 * 10^{-2}$ for all parameters. We stop the $\epsilon$-4DVar method after 25 iterations, which already represents 1450 model simulations, as it seems to show no significant improvement in the minimisation of its cost function. We find that 4DEnVar gives a mean reduction in RMSD of $94.3\%$ across all stations with a maximum reduction of RMSD at the South African station, Cape Point (CPT) ($98.8\%$ ) and minimum RMSD reduction at the Finland station, Pallas (PAL) ($85\%$). The $\epsilon$-4DVar gives a mean reduction in RMSD of $92.5\%$ across all stations with a maximum reduction of RMSD at the Chinese station, Walinguan (WLG) ($96.9\%$ ) and a minimum RMSD reduction at the Australian station, Cape Grim (CGO) ($81.3\%$). The average RMSD drops from $3.35$ ppm to $0.17$ ppm after assimilation for 4DEnVar and to $0.24$ ppm for $\epsilon$-4DVar. We calculate the MAD score between the "true" parameter and the prior/posterior parameters after normalising between 0 and 1 (because the parameters do not have the same units). This normalisation allows us to bound the MAD score between 0 and 1. The normalised MAD score between the "true" parameters and the prior parameters is $0.17$. After assimilation using 4DEnVar, a $53\%$ reduction in this score is obtained, giving a normalised MAD score of $0.08$. The $\epsilon$-4DVar method gives a reduction in the normalised MAD of $15\%$ giving a normalised MAD score of $0.14$. Fig. 6 shows the prior, "true" and posterior parameter values obtained using both methods. For each parameter, we calculate the MAD score independently. The 4DEnVar method gives a MAD reduction of $44.7\%$ for $\boldsymbol{Vcmax}$, $78.2\%$ for $\boldsymbol{SLA}$, $36.3\%$ for $\boldsymbol{LAI_{max}}$, $54.2\%$ for $\boldsymbol{m_{maint.resp}}$ and a reduction of the absolute difference (AD) of $98.8\%$ for $\boldsymbol{Q10}$. The $\epsilon$-4DVar method gives a MAD reduction of $11.3\%$ for $\boldsymbol{Vcmax}$, $32.7\%$ for $\boldsymbol{SLA}$, $9.6\%$ for $\boldsymbol{LAI_{max}}$, $4.2\%$ for $\boldsymbol{m_{maint.resp}}$ and a reduction of the AD of $97.5\%$ for $\boldsymbol{Q10}$. Fig. 7 illustrates the spatial disparities in net land carbon fluxes between the "synthetic" fluxes and the prior/posterior estimation of the two methodologies, in addition to their mean annual global net carbon flux. The 4DEnVar method achieved a mean annual global net flux of $-2.62$ GtC/year, with a difference of $0.05$ GtC/year compared to the "synthetic" fluxes. Spatial differences were limited to an absolute maximum of $0.28$ gC/m$^2$/day, with an absolute mean of $0.009$ gC/m$^2$/day. In contrast, the $\epsilon$-4DVar method produced a mean annual global net flux of $-2.43$ GtC/year, with a difference





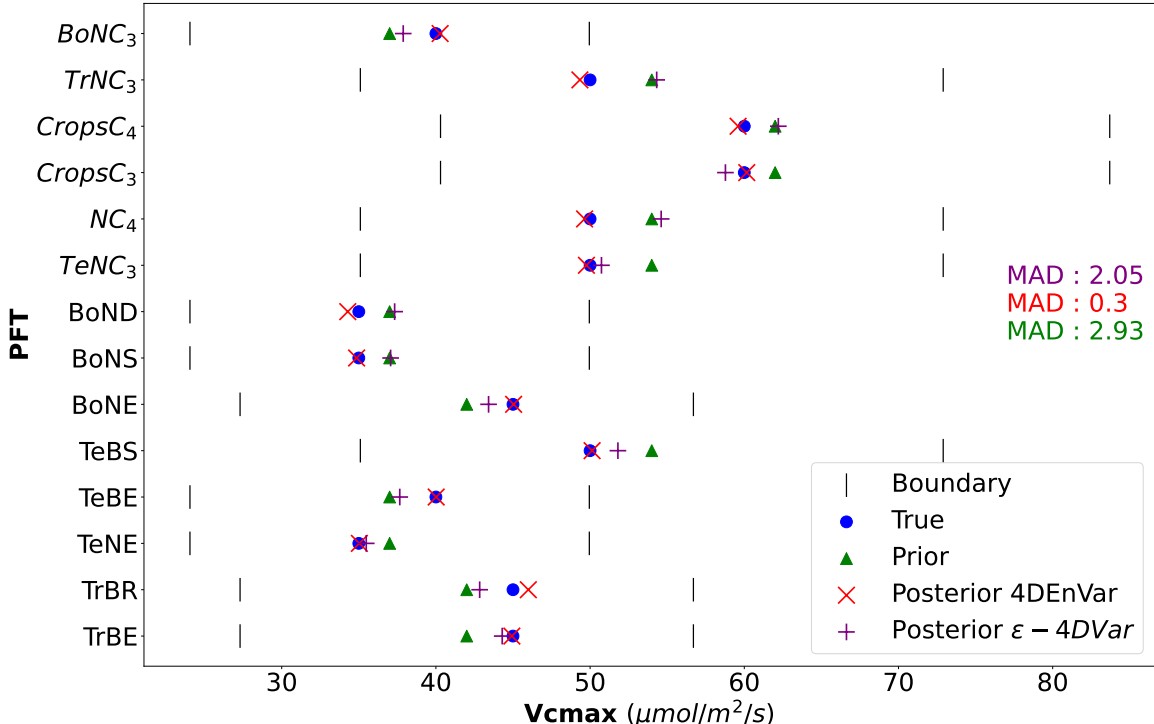

**Figure 3.** Result in parameter space for the simple case: the prior parameter values are represented by the green triangle and the posterior parameter values after optimization are represented by the purple + symbol for the $\epsilon$-4DVar method and the red × symbol for the 4DEnVar method. The blue circles represent the "true" values used to produce the assimilated 'synthetic' observations. The Mean Absolute Difference (MAD) score shown is calculated between the "true" parameter values used to generate the "synthetic" observations and the different parameter values following the same color code (green score using prior parameter, purple score using posterior parameter of $\epsilon$-4DVar, red score posterior parameter of 4DEnVar).

of $0.24$ GtC/year relative to the "synthetic" fluxes. Spatial differences for this method reached an absolute maximum of $0.6$ gC/m$^2$/day, with an absolute mean of $0.031$ gC/m$^2$/day.

## 4  Discussion

### 4.1  Experiments

In Sect. 3.2.1, we found that the 4DEnVar method outperforms the $\epsilon$-4DVar method, both in terms of RMSD reduction and
MAD score, and with a smaller number of model simulations for the simple case. The 4DEnVar method reduces the RMSD by $97\%$, and almost recovers the "true" parameters, whereas the $\epsilon$-4DVar method reduces the RMSD by $82\%$ and seems to





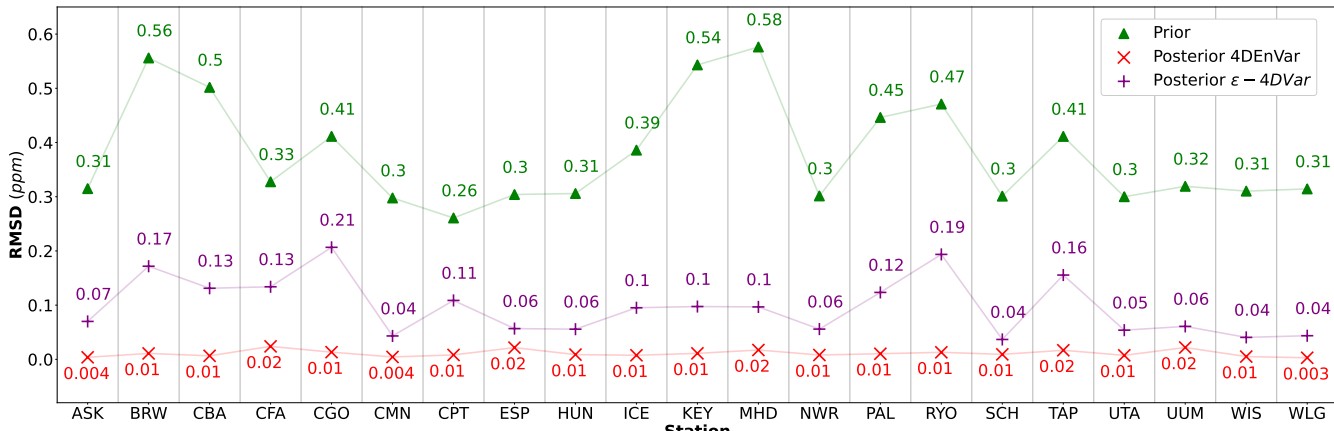

**Figure 4.** The Root Mean Squared Differences (RMSD) scores between synthetic observations and prior simulations for the simple case for each of the 21 atmospheric stations are represented by green triangles. The RMSD scores between synthetic observations and posterior simulations given by the $\epsilon$-4DVar (4DEnVar) method are represented by the purple $+$ symbol ( red $\times$ symbol)

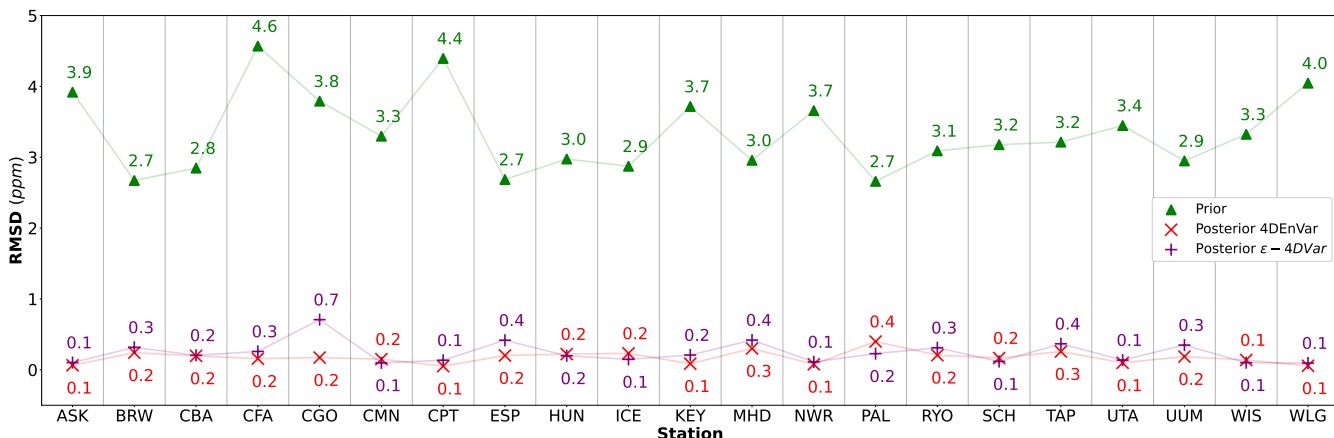

**Figure 5.** The Root Mean Squared Differences (RMSD) scores between synthetic observations and prior simulations for each of the 21 atmospheric stations for the complex case are represented by green triangles. The RMSD scores between synthetic observations and posterior simulations given by the $\epsilon$-4DVar (4DEnVar) method are represented by the purple $+$ symbol ( red $\times$ symbol)

converge into a local minimum. In addition, 4DEnVar requires three times fewer simulations. Other configurations presented in Section 3.1 show that the 4DEnVar, using 50 members, leads to similar RMSD reduction as $\epsilon$-4DVar (see Table 1). However, this 4DEnVar configuration still gives a better MAD score of 0.44 giving a reduction of the prior MAD by 85% - this shows that 4DEnVar method is less influenced by local minima than the $\epsilon$-4DVar method. We can also note that using the $\epsilon$-4DVar method results in *a posterior* parameter values that either i) remain close to the *a priori* values or that ii) increase the distance






**Figure 6.** Results in parameter space for the complex case: the prior parameter values are represented by the green triangle and the posterior parameter values after optimization are represented by the purple + symbol for the $\epsilon$-4DVar method and the red $\times$ symbol for the 4DEnVar method. The blue circles represent the values used to produce the assimilated 'synthetic' observations. The MAD (or the absolute differences for **Q10**) scores shown are calculated for each parameter independently.

from the value of the "true" parameters. The first case can be explained by the lower sensitivity of the parameters concerned. The sensitivity of the **Vcmax** parameter depends on the associated PFT. Not all PFTs have the same impact on NBP fluxes, as they do not have the same spatial distribution or the same proportion. (see Table A1). This is the case for the parameters

of PFT TeBE, BoNS, BoND, CropsC$_4$ and TrNC$_3$ which have a proportion equal to or less than 3% , and are therefore less influential on global NBP fluxes. The second case can be explained by self-compensation due to the equifinality of the problem. Indeed, as some parameters are not properly calibrated, others compensate and may not converge towards the "true" minimum.

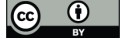


It may concern the parameter of PFT $NC_4$. The 4DEnVar method seems to be less affected by these problems and is therefore a promising solution.

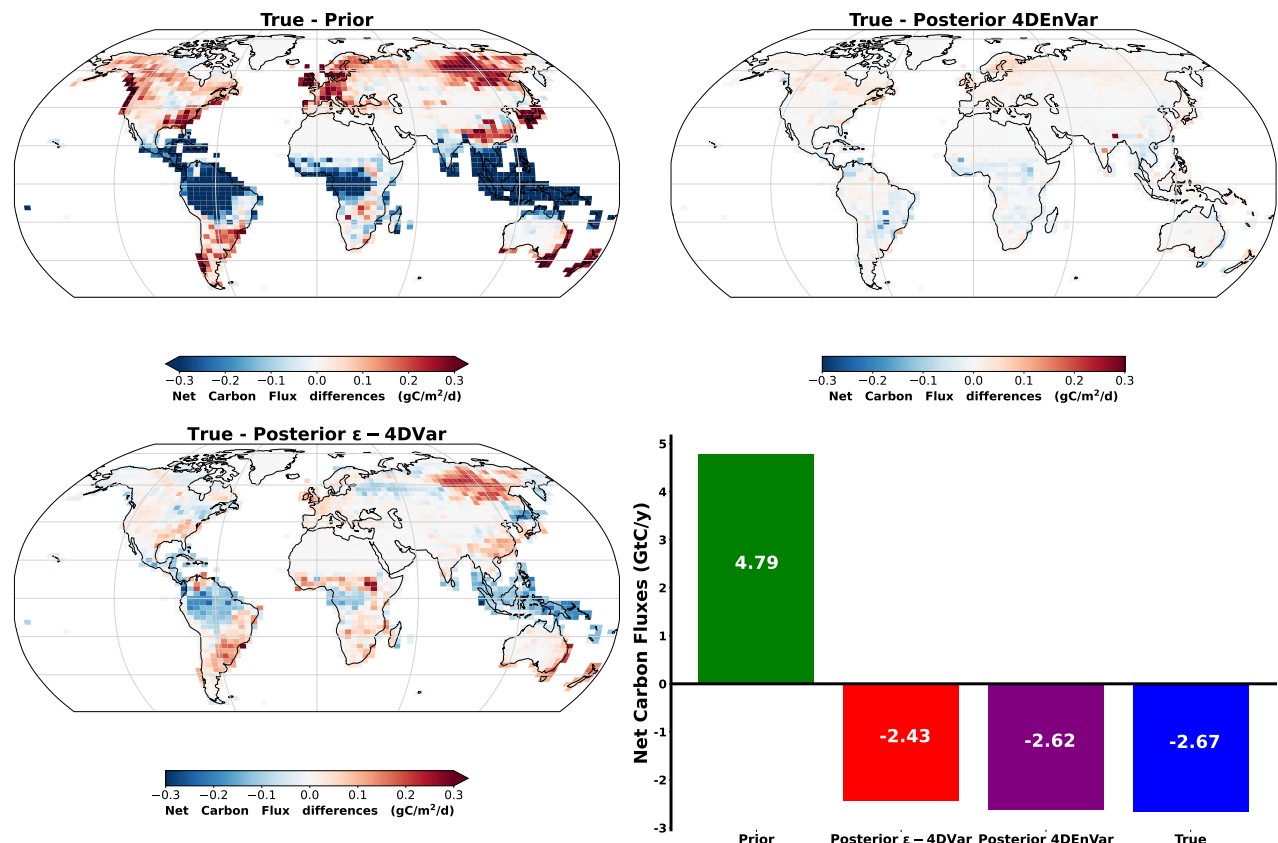

**Figure 7.** Spatial differences in net land carbon fluxes between "synthetic" fluxes and the prior/posterior estimate of the two methods alongside their mean annual global net carbon flux for the complex case. (Negative values are carbon uptakes and positive values are carbon emissions)

In Sect. 3.2.2, we have seen that the 4DEnVar method is able to calibrate 57 parameters and reduces the mean RMSD by 94.3%, which is slightly better that the $\epsilon$-4DVar method with a mean RMSD reduction of 92.5%. It is worth noting that the mean RMSD reduction may be better in the complex case than in the simple case . This is mainly due to the fact that the *a priori* error is much larger in the complex case. Nevertheless the mean RMSD score remains bigger than in the simple case (0.17 ppm and 0.24 ppm for the complex case and 0.01 ppm and 0.1ppm for the simple case for 4DEnVar and $\epsilon$-4DVar respectively) .

While the mean RSMD reduction scores are similar for the complex case, the MAD scores in parameter space are different. In fact, the 4DEnVar method is closer to the "true" parameters by reducing the normalised MAD by 53%, whereas the $\epsilon$-4DVar method remains very close to the *a priori* position. Both methods are capable of recovering the "true" **Q10** parameter since it is the most sensitive parameter. The $\epsilon$-4DVar method seems to have difficulties in the calibration of the parameter $m_{maint.resp}$





and **LAI** showing reductions in MAD scores that are less than $10\%$. Considering Fig. A2, we can see that some PFTs give a
partial derivative that do not completely converge with an $\epsilon$ of $10^{-2}$ (for example the PFT CropsC$_4$), so it is likely that the $\epsilon$
for these parameters are underestimated. Other PFTs seem to give a partial derivative that do completely converge with an $\epsilon$ of
$10^{-2}$ (for example the PFT TrBE), but remain close to their *a priori* value, so it is likely that the sensitivity of these parameters
is low. The other parameters are therefore self-compensating and this may partly explain the poorer performance of this method
in terms of MAD score which are always better for the 4DEnVar method. The self-compensating effect can be illustrated in Fig.
7. The posterior spatial distributions of net carbon flux obtained from the two methods exhibit notable differences. It appears
that the $\epsilon$-4DVar method obtains a different spatial structure of the net carbon fluxes. Indeed, the carbon fluxes absent from one
region can be reallocated to another, resulting in only minor variation in atmospheric $CO_2$ concentration. It is therefore notable
that the 4DEnVar method demonstrates superior performance, as it is more aligned with the 'synthetic' net carbon fluxes
both spatially and globally than the $\epsilon$-4DVar method. However, the 4DEnVar method outperfoms the $\epsilon$-4DVar method also
in terms of computational cost: the 4DEnVar method only needs 300 simulations, whereas the $\epsilon$-4DVar method needs 1450,
which means a reduction in computing time of almost five times. This experiment of calibrating a large number of parameters
represents a more realistic case, even if we do not consider the model/observation error. The use of 4DEnVar here therefore
demonstrates its ability to calibrate many parameters with fewer model simulations. In this experiment, one simulation took 11
minutes (wall times) on average, using 20 CPUs of a computer server (using Intel Xeon Gold 5115 processor). Neglecting the
other computational times, using the $\epsilon$-4DVar for the complex case represents more than 265 hours of computation, where it
only represents 55 hours for the 4DEnVar. Such a reduction cannot be ignored since a simulation in this experiment represents
a short (only 2 years), low-cost model configuration - low ORCHIDEE spatial resolution and use of pre-calculated LMDZ
transport fields.

     The poorer performance of the $\epsilon$-4DVar method is likely related to inaccurate determination of $\epsilon$, which results in inaccurate
estimates of the tangent linear and adjoint models. The 4DEnVar method avoids the development and maintenance of tangent
linear and adjoint models, and ensures a fully functional assimilation method that does not require the use of finite differences.
It seems that the subjective choice of the 4DEnVar set-up, i.e. the size and distribution of the ensemble, is less critical than
the subjective choice of $\epsilon$ used in the $\epsilon$-4DVar, which must be determined independently for each of the parameters and given
assimilated data-streams (with the associated number of model simulations). Moreover, like the tangent linear and adjoint
models, this $\epsilon$ must be re-tuned for a different model as the sensitivity of the parameter can be different. Indeed, other studies
using different versions of the ORCHIDEE LSM have used different $\epsilon$ values (Santaren et al., 2007; Kuppel et al., 2012; Peylin
et al., 2016; Bacour et al., 2023).

     The results obtained here for the $\epsilon$-4DVar are not equivalent to a standard 4Dvar using a tangent linear and adjoint model.
Therefore, we can draw no conclusions on the comparison between the 4DEnvar and standard 4DVar methods as was high-
lighted in Liu et al. (2008). A potential - but hard to implement - way to improve the $\epsilon$-4DVar may be to have a dynamic $\epsilon$ that
becomes more refined as the methods converge. Nevertheless, even with the "perfect" $\epsilon$, we cannot guarantee that the $\epsilon$-4DVar
method would be less computationally expensive.





## 4.2 Challenges and perspectives

This study relies on twin experiments, which eliminate the complexities associated with model/observation errors, and allows us
to focus on the performance of two assimilation methods. This experiment highlights the superiority of the 4DEnVar method to
assimilate atmospheric $CO_2$ concentration data. However, the assimilation of real observations is not straightforward. The use
of real data must be followed by characterisation of the model/observation errors. If the model/observation errors are incorrect,
the 4DEnVar method can give infeasible *a posterior* parameter values, i.e. outside the imposed parameter boundaries (and
therefore give non-physical parameter values). Furthermore, even with feasible *a posterior* parameter values, the parameters
obtained may be beyond the assumption of linearity made by the use of linear combinations in Eq. 17 and therefore do
not improve the associated simulation. Nevertheless, several techniques seem promising for managing these limitations. The
inclusion of a weight factor in the background term, as is done in (Raoult et al., 2016), and a better definition of the error
covariance matrix $\mathbf{B}$ may provide a solution. Some of these challenges are not specific to the 4DEnVar method and are common
to the 4DVar method (Raoult et al., 2024b). These challenges are therefore the subject of active research to improve the
assimilation of real observational data.

This study acts as a proof-of-concept for the assimilation of atmospheric $CO_2$ concentration data using adjoint-free methods.
The next steps for the future would be to use real observations, which come with other technical and scientific problems (e.g.
quantifying the model/observation error). Future studies should focus on the assimilation of more recent and more spatially
distributed atmospheric $CO_2$ concentration data - e.g. satellite $XCO_2$ product, using 4DEnVar. To do so, a more recent version
of LMDZ and/or ORCHIDEE should be used. Those studies will focus on the processes involved in the carbon cycle to improve
their parameterisation and/or to detect any missing processes in the model. As the 4DEnVar method only requires forward
simulations of the models, it is easy to change the model (either the LSM or the atmospheric transport model). Furthermore,
the method is easy to parallelise as each element of the ensemble is independent. Once built, no further call of the model is
necessary (except in the analysis step), which allows us to explore different configurations, e.g. in the construction of the error
matrix $\mathbf{R}$ or the weighting of background terms - both of which play a key role in the assimilation of real observations - without
additional computational cost.

Despite extensive research on the automatic generation of tangent linear and adjoint models - either using new languages or
differential software - it remains an enormous challenge to acquire and maintain tangent linear and adjoint models for complex
and continuously evolving models. However, it is still a key priority to understand structural errors, to quantify uncertainties
and to refine future predictions via parameter calibration. The use of adjoint-free data assimilation methods such as 4DEnVar
is therefore an excellent opportunity, as it can be implemented quickly and requires no model modification.

Moreover, the 4DEnVar method can also be used to assimilate other types of observations and calibrate parameters involved
other processes. Adding different observation terms (one term per data stream) to the cost function will also enable multi-data
stream assimilation. This has already been used to assimilate several types of data using the JULES LSM (Pinnington et al.,
2020, 2021; Cooper et al., 2021), which also demonstrates its ability to be model-independent.





# 5    Conclusions

We have shown that the 4DEnVar method has great potential for calibrating ORCHIDEE parameters assimilating atmospheric $CO_2$ concentration data and using the LMDZ atmospheric transport model. The method was tested on a so-called *twin exper-iment* using two different cases: 1) a simple case where 4DEnVar effectively recovered the "true" parameter values, whereas

the $\epsilon$-4DVar method, despite reducing the RMSD, failed to recover the "true" parameters; and 2) a complex case where both methods achieved up to a 90% reduction in RMSD, with 4DEnVar showing slightly better performance, including a lower MAD score in parameter space, indicating greater efficiency in parameter recovery and an improved alignment with 'synthetic' net carbon fluxes, both spatially and globally. Additionally, 4DEnVar is computationally less demanding and does not require the development or maintenance of tangent linear and adjoint models, facilitating the use of updated model versions without

modification. By successfully applying this method to the ORCHIDEE model with a pre-calculated LMDZ transport model, we have illustrated its adaptability, making it well-suited for other land surface models, whether coupled with atmospheric transport models or not.

*Code and data availability.*   The source code for the ORCHIDEE version used in this model is freely available online at https://doi.org/10.14768/c68bc728-da71-4383-84df-dcde31d9c006 ORCHIDEE (2025). The ORCHIDAS 4DEnVar code and data used in this paper are available from a Zenodo

repository at https://doi.org/10.5281/zenodo.14609416 Beylat (2025).

# Appendix A

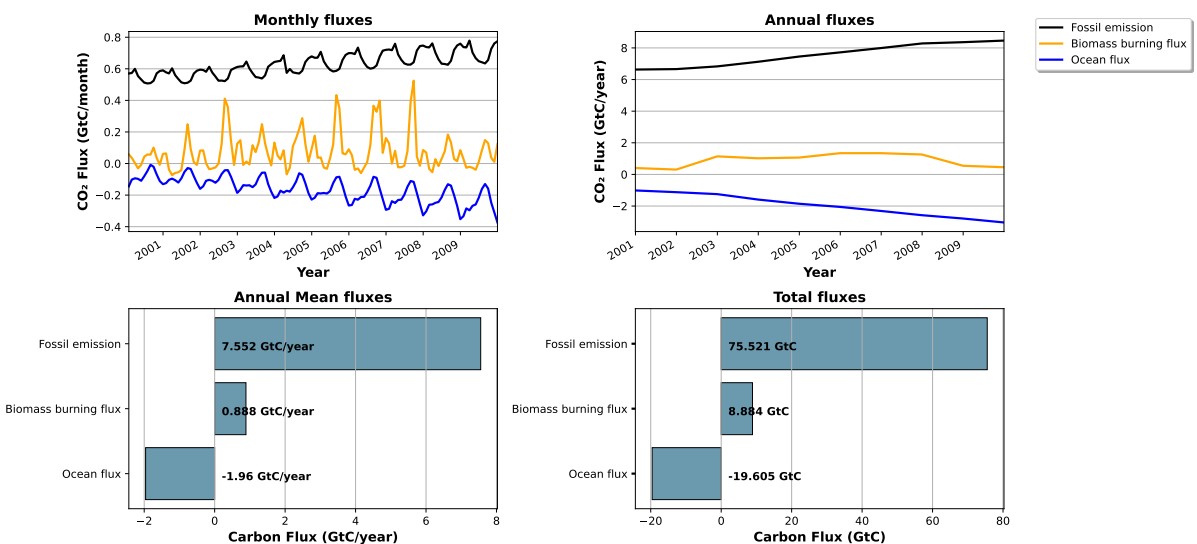

**Figure A1.** Ocean, Biomass burning Carbon Fluxes and Fossil emission (2000-2009)



**Figure A2.** $\epsilon$-test: Mean of partial derivative as a function of $\epsilon$. The partial derivative of the $\mathcal{H}$ model is calculated with respect to the parameters use in complex case for each PFT. It is calculated on the concentration space using every station over 2 years. The mean of the partial derivative is then calculated over space and time in order to visualise the local derivative. The derivative of $\mathcal{H}$ is calculated for several $\epsilon$.

*Author contributions.* SB and PP conceived on the study. VB implemented the coupling of ORCHIDEE and LMDZ, originally developed by PP. SB implemented the 4DEnVar method and VB implemented the $\epsilon$-4DVar method. SB performed and analysed the DA experiments. NR, PR, PP, and CB provided expertise on the assimilation of Atmospheric $CO_2$ concentration and the $\epsilon$-4DVar method. ND and TQ provided expertise on the 4DEnVar method. SB prepared the article with contributions from all co-authors.





**Table A1.** Plant functional types (PFT) in ORCHIDEE and acronyms used in this study.

| PFT | Acronym | Proportion |
|---|---|---|
| Tropical Broadleaf Evergreen | TrBE | 10.3% |
| Tropical Broadleaf Raingreen | TrBR | 5.92% |
| Temperate Needleleaf Evergreen | TeNE | 4.51% |
| Temperate Broadleaf Evergreen | TeBE | 2.31% |
| Temperate Broadleaf Summergreen | TeBS | 5.09% |
| Boreal Needleleaf Evergreen | BoNE | 4.72% |
| Boreal Broadleaf Summergreen | BoNS | 2.29% |
| Boreal Needleleaf Deciduous | BoND | 2.59% |
| Temperate Natural Grassland (C3) | $TeNC_3$ | 6.43% |
| Natural Grassland (C4) | $NC_4$ | 8.73% |
| Crops (C3) | $CropsC_3$ | 11.4% |
| Crops (C4) | $CropsC_4$ | 3.04% |
| Tropical Natural Grassland (C3) | $TrNC_3$ | 1.58% |
| Boreal Natural Grassland (C3) | $BoNC_3$ | 10.6% |

**Table A2.** Mean of the partial derivative for all parameter for each PFT computed using $\epsilon$ allowing a stable derivation. The partial derivative is calculated on the concentration space using every station over 2 years. The mean of the partial derivative is then calculated over space and time

| Parameters | $Vcmax$ | $SLA$ | $LAI_{max}$ | $m_{maint.resp}$ | | Q10 |
|---|---|---|---|---|---|---|
| $\epsilon$ | $10^{-3}$ | $10^{-3}$ | $10^{-2}$ | $10^{-2}$ | | $10^{-3}$ |
| **PFT** | | | | | **Global** | -12.7 |
| TrBE | $-0.153$ | $-63.7$ | $-0.07$ | $104.7$ | | |
| TrBR | $-0.042$ | $-38.7$ | $0.03$ | $17.2$ | | |
| TeNE | $-0.077$ | $-103.3$ | $-0.02$ | $12.1$ | | |
| TeBE | $-0.06$ | $-15.1$ | $-0.13$ | $11.4$ | | |
| TeBS | $-0.04$ | $-30.6$ | $-0.06$ | $8.5$ | | |
| BoNE | $-0.057$ | $-126.2$ | $-0.19$ | $13.8$ | | |
| BoNS | $-0.065$ | $-44.6$ | $-0.07$ | $5.4$ | | |
| BoND | $-0.023$ | $-23.6$ | $-5*10^{-5}$ | $1.5$ | | |
| $TeNC_3$ | $-0.053$ | $-44.9$ | $-0.43$ | $7.9$ | | |
| $NC_4$ | $-0.072$ | $-27.0$ | $-0.22$ | $12.0$ | | |
| $CropsC_3$ | $-0.072$ | $-89.3$ | $-0.15$ | $16.4$ | | |
| $CropsC_4$ | $-0.023$ | $-18.3$ | $-0.07$ | $3.3$ | | |
| $TrNC_3$ | $-0.017$ | $-23.0$ | $-0.01$ | $3.1$ | | |
| $BoNC_3$ | $-0.056$ | $-40.7$ | $-0.28$ | $5.7$ | | |



*Competing interests.* The authors declare that they have no competing interests.

*Acknowledgements.* This work has been supported by a scholarship from CNRS under the Melbourne–CNRS joint doctoral programme.



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
