# Peer review of "Towards the Assimilation of Atmospheric CO2 Concentration Data in a Land Surface Model using Adjoint-free Variational Methods"

_EGUsphere, 2025_

## Referee Comment (RC2)

While this manuscript addresses a highly relevant and important topic—adjoint-free data assimilation for land surface model parameter estimation using atmospheric $CO_2$ concentrations, there is a fundamental conceptual misunderstanding regarding the data assimilation framework employed. The authors repeatedly describe their method as a "4DEnVar" approach. However, after careful review of the methodology and experimental design, it is clear that the implemented framework aligns more closely with a 3DEnVar method rather than a true 4DEnVar. I will provide more specific comments below detailing the evidence for this classification error and offering suggestions for how to appropriately revise the manuscript.

1. The authors incorrectly label their method as "4DEnVar." In classical data assimilation terminology, the key distinction between 3D and 4D variational methods lies in the incorporation of assimilation windows. A three-dimensional variational (3DVar or 3DEnVar) method assimilates observations as a function of space only, without explicitly considering the time evolution of the model states. In contrast, a four-dimensional variational (4DVar or 4DEnVar) method introduces a temporal dimension by defining an assimilation window, allowing the model to evolve dynamically and best fit the observations distributed across time within that assimilation window. In this manuscript, there is no explicit mention of an assimilation window, nor is there any evidence that the model trajectory evolves and interacts with observations at multiple times during a window. Instead, observations appear to be treated statically, consistent with a 3DEnVar framework. Therefore, the method used in this study should be accurately referred to as 3DEnVar, not 4DEnVar. The manuscript must be revised to correct this misclassification throughout, including the title, abstract, methods, results, and discussion sections.

2. The description at L56–58 defines general variational assimilation, not 4DVar specifically. 4DVar uniquely involves an assimilation window and time-evolving model trajectory. Please correct this definition.

3. Data assimilation can generally be categorized into full-field assimilation and anomaly assimilation. Full-field assimilation adjusts the model state towards observed absolute values, while anomaly assimilation only incorporates the observed anomalies. The authors must clearly specify which approach is used. If full-field assimilation is applied, the impact on climatology should be explicitly evaluated and presented.

4. The cost function shown in Equation (5) corresponds to the standard 3DVar formulation, as it lacks any temporal dimension or model trajectory integration. A true 4DVar cost function should involve the evolution of the model state over time:

$$J(x) = \frac{1}{2}(x - x_b)^T B^{-1}(x - x_b) + \frac{1}{2}\sum_{i=0}^{n}(H_i M_{t_0 \to t_i} x - y_{obs}^i)^T R_i^{-1}(H_i M_{t_0 \to t_i} x - y_{obs}^i)$$

Please revise both the equation and the method name accordingly.

5. Lines 310-315: The R matrix represents the observation error covariance (not both the model/observation error) and should account for spatial heterogeneity in observation uncertainty. Setting all diagonal elements to 0.01 ppm is overly simplistic. Even in a simple setup, R could be statistically computed based on the variance of the observation. Please consider a more realistic design or justify this simplification.

6. Lines 364-366: Please clearly define how RMSD is computed, and indicate whether systematic bias is removed prior to calculation.

7. The evaluation relies almost entirely on RMSD and MAD. Please consider providing additional diagnostic metrics, such as correlation coefficients between posterior and true fields, or skill scores, to offer a more complete picture of assimilation performance.

8. The differences in RMSD reductions between different methods and configurations are discussed, but no statistical tests are provided. Please include simple significance tests (e.g., paired t-tests) to assess whether the differences in RMSD reductions are statistically meaningful across stations.

9. The comparison between 4DEnVar and ε-4DVar is repeatedly emphasized, but ε-4DVar is not equivalent to standard 4DVar. Please emphasize earlier and more clearly

that the ε-4DVar results are only a rough approximation and that conclusions should not be generalized to comparisons with a full 4DVar system.

10. All evaluations are performed against the same synthetic dataset used for assimilation. For robustness, a portion of the synthetic observations should be withheld during assimilation and used for independent validation.

---

## Author Comment (AC1)

**Review #1**

Evaluating the overall quality ("general comments"),

The authors conduct a twin model experiment to test the ability of two competing parameter estimation techniques (eta-4DVAR vs. 4DEnVar) to constrain 54 parameters within the land model ORCHIDEE.  The authors use synthetic observations to generate atmospheric CO2 station data in which to retrieve the 'true' parameters from prior parameter distributions.   The authors find that the 4DEnVar approach performs the best in terms of the RMSE statistic of global NBP and in terms of the posterior parameter values relative to the true values.   The authors claim this demonstrates strong potential for 4DEnVar to be used with real data, and should be widely applicable to other land surface models.

We would like to thank the reviewer for taking the time to read through and comment on this manuscript, comments that will strengthen the paper and help clarify key points of the paper.

This reviewer found the topic relevant to the current state of earth system science which requires a wide range of observations to calibrate model performance to improve forecasts/projections.   The potential for such a problem to be ill-constrained and suffer from equifinality stood out to this reviewer given the use of a single data set (atmospheric CO2) to constrain a multi-dimensional problem. This author recommends a more nuanced discussion of equifinality for this application both in the twin model experiment and for potential applications of using real data (see scientific questions below).  This reviewer also would have appreciated a better description of how the parameter values were sampled (perturbed) from their respective distributions – and also to what extent the authors could have presented their results for the 4DEnVar using parameter distributions rather than point estimates.  It was somewhat surprising how well the posterior parameter values improved the global distribution of NBP, given the limited range of the CO2 station footprint.  More discussion related to the distribution of land PFTs relative to the CO2 station footprint may have been helpful to aid this discussion.

We thank the reviewer for highlighting these very important points, which we have addressed by improving existing figures, adding a new one, and revising or expanding the text accordingly. Please see the specific changes in the manuscript, as detailed in our response under each reviewer's comments. We agree that the

inherent equifinality of such data assimilation problems was not properly addressed in the discussion. We have added a paragraph to the discussion to address this issue and believe that this comment strengthens the article.

Individual scientific questions/issues ("specific comments")

(Please note that changes to the text are referenced by the line number in the new manuscript.)

Lines 45-50:   If you calibrate against CO2 observations only, and do not adjust for model biases in land carbon pools – how accurate can your projections/forecasts of the carbon cycle be?

Thank you for raising this important point. While it is true that calibrating solely against atmospheric CO2 observations may not fully address biases in land carbon pools, it still provides a valuable global constraint on carbon fluxes. This approach ensures that the overall carbon budget is consistent with observed atmospheric CO2 concentrations, which is crucial for accurate projections and forecasts of the carbon cycle. To further improve the accuracy of our model, we acknowledge the importance of incorporating additional data sources integrating remote sensing data and in-situ measurements of land carbon pools to better constrain and reduce model biases. This multi-faceted approach would enhance the reliability of our projections and provide a more comprehensive understanding of the carbon cycle dynamics.

In this study, we assume that performing a complete TRENDY simulation, as described in Section 2.1.1, provides a realistic estimate of carbon sinks. However, we would like to emphasise that the main focus of this article is methodological - we present a two-year twin experiment designed to evaluate the feasibility and performance of the assimilation framework using atmospheric $CO_2$ data.

In the case of assimilation of real observations, the ideal solution would be also to include a pre-assimilation phase in the assimilation, i.e. the complete spin-up + transient simulation, to monitor the  effect of the initialisation of the land carbon pools in the atmospheric $CO_2$ concentration. However, given the cost of such simulations, this may not be feasible. Another potential solution is to run a small transient simulation with the modified parameter before the assimilation windows. For example, when assimilating the year 2000-2005, start the simulation in 1995

until 2005 and only consider the last five years (2000-2005). This way, the carbon reservoir would be modified and influenced by the parameter. It would also allow for better monitoring of the effect of long-term changes and thus increase confidence in the projection.

Line 95: "We demonstrate the potential of 4DEnVar using synthetic observational data and compare its performance with that of 4DVar with finite differences."

Your criteria for testing the differences between the methods is vaguely stated here. Are you judging success based on which method best identifies the 'true parameters'.

We added more detail in L96:

> We demonstrate the potential of 4DEnVar using synthetic observation data **according to different criteria: i) the differences between synthetic observation and simulation of atmospheric CO2 concentration, ii) the spatial distribution of carbon flux as well as their subcomponent, and iii) the recovery of the true parameters used to generate the synthetic observation. We also compare the performance of 4DEnVar using these criteria** with that of 4DVar with finite differences

Lines 135:140: Given the use of pre-calculated transport fields that relate atmospheric concentrations to surface fluxes, you do not make use of a dynamic atmospheric ensemble to generate this relationship based upon actual atmospheric forcing. This seems a bit like using a background climatology to get the concentration/surface flux relation. Also the land seems to be decoupled from your atmosphere – in the sense the dynamics that drive the CO2 concentration/flux relationship is not the same as the actual met forcing driving the land model. This is perhaps not as important given the authors are generating 'perfect' obs, but during implementation for real parameter estimation should this not have an important impact?

The land model is decoupled from the atmosphere, and ORCHIDEE is run offline using meteorological forcing files from a reanalysis of ERA-Interim data every 3 hours. The use of reanalysis meteorological fields is important in order to better respect the temporal dependence of meteorological events. The pre-calculated transport field is generated using the LMDZ model, which is driven by winds from

the same reanalysis meteorological data. This is not the same as using climatological data, as the calculation is performed using reanalysis winds for the same years as those of the simulations. The use of these precalculated transport fields reduces the calculation time when running the same atmospheric simulation several times by only changing the surface fluxes. We add more clarification by adding in L141:

> **The calculated winds ( u and v) used to drive LMDZ are provided by ERA-Interim reanalysis meteorological data in order to realistically account for the temporal dependence of meteorological events.**

and L151

> **Nevertheless, it is important to note that the use of these pre-calculated transport fields does not allow for the evaluation of dynamic feedbacks between the surface and the atmosphere that may occur due to parameter changes.**

Line 150-155, Figure 1:  One would expect that the $CO_2$ sites were also chosen such that land surface areas sensitive to atmospheric $CO_2$ also coincide with the range of land surface PFT types in this analysis.  Any consideration of this?

Line 280-85 – Same question as before, these sites were chosen to be sensitive to land surface fluxes, however, do they provide good sampling of the most important PFTs?  Sampling of North America and South America look poor.   Sampling of Africa does not include the tropics at all…..

In this study, the authors considered the pre-calculated transport fields for a handful of actual atmospheric stations (21). The locations of the stations are therefore determined by the actual observation networks. It is true that some PFTs are undersampled as there are more stations in northern latitudes, mainly in America and Europe, than in the rest of the world.  We selected the stations based on their sensitivity to continental fluxes. Although the number of stations is not ideal, it allows us to monitor terrestrial carbon fluxes. The other stations were mainly located over the ocean and would have been less affected by changes in the terrestrial component given the chosen assimilation window. Only the TrBE (Tropical Broadleaf Evergreen forests) PFT, mainly found in the Amazon rainforest and

Central Africa, and the BoND (Boreal Deciduous Needleleaf forests) , mainly found in Siberia, appear to be less observed. We believe that this is also related to the results presented in Figure 7 for Epsilon 4Dvar. Indeed, we see from the well distribution that the largest differences appear in these two regions, which could be explained by the fact that they are less monitored.

We have added the following text to the manuscript L162

> **The selected stations also provide a good overview of most PFTs. However, as we can see in Fig. 1, two PFTs appear to be less sensitive to the selected stations: TrBE, which is mainly found in the tropical forests of Amazonia and Central Africa, and BoND, which is mainly found in Siberia.**

L486

> **We believe that the different spatial structure obtained by e-4DVar against the synthetic net carbon flux could be explained by the fact that the two PFTs TrBE and BoND are not well monitored, creating a dipole in the Amazonian and Siberian regions to compensate for the incorrect carbon flux corrections in other regions.**

Line 286:  How was your prior parameter distributions chosen?  From the uniform distribution shown in the figures where you only show upper and lower bounds? Or from a normal distribution as described in Equation 14 and 15?

The new a priori parameter vector was manually perturbed in order to find a new set of parameters giving a simulation of the a priori atmospheric concentration consistent with the observations or simply realistic. We added to the text L296:

> **A new vector of a priori parameters was generated manually, ensuring that it differed from the "real" parameter values while retaining physically meaningful values.**

Figure 3:  Showing the ensemble mean parameter behavior  doesn't give any information on the ensemble distribution.  Maybe I am misinterpreting the implementation of the 4DEnVar method, but can't you show this in terms of the true, prior and posterior *distributions* instead of the ensemble mean behavior?

Figure 6: Same question as before – can you convey this information in terms of distributions (histograms)?

Figures 3 and 6 have been modified to display the standard deviation of the posterior ensemble as an error bar that can be considered as the posterior uncertainty. We prefer not to use histograms, as all distributions are Gaussian. The mean and standard deviation are therefore sufficient to visualise the ensemble. The use of histograms is very useful, but would make the figure difficult to read, particularly in the complex case involving 57 parameters (which would result in 57 histograms). The legends have been modified as well.

We added in L274:

**A posterior ensemble can be generated as it is described by Douglas et al 2025 by calculating Xa' where**

$$X'_a = X'_b(I + (HX'_b)^T R^{-1} - 1HX'_b)^{-\frac{1}{2}}$$

and in L463:

**Furthermore, Fig. 3 shows a significant decrease in the standard deviation of the posterior ensemble. This allows us to identify which parameters and therefore which PFTs appear more sensitive. In this case, it seems that the results for the TrNC3 and Crops C4 PFTs are the most uncertain.**

and in L473:

**The posterior ensemble generated for the 4DEnVar also shows a reduction in uncertainty for all parameters. This uncertainty reduction is not equal for all parameters - a maximum reduction can be seen for the Q10 parameter(reducing the standard deviation by 94 %) and the lowest for the less sensitive $m_{maint.resp}$ parameter (with a 14% reduction for the NC4 PFT).**

Figure 6: (SLA panel) Why do most of the prior values for the parameters all start at the same value? Were they not being perturbed independently?

Indeed, the parameters were not perturbed  independently; the new set of *a priori* values used in data assimilation was derived from the  'actual' *a priori* values of the model which may be identical across different PFTs. We applied the same perturbation for these PFTs.

Table A1:   What does proportion mean in this context?

Line 430:   As far as I can tell, it is still not defined what the 'proportion' of the parameters is.  Is it based on global land area coverage, or land coverage that coincides with spatial footprints from your chosen network?  These could be two completely different things.  Did you do any comparison of the MAD statistic based on % of land area covered by the CO2 network spatial footprint?  Are they strongly related?  Would be nice to see a land surface map with PFT distribution.

Plant functional types (PFT) in ORCHIDEE and acronyms used in this study as well as their proportion

The proportion  is the Global Cover Fraction by the PFT, the remaining proportion being Bare Soil. The title of the Table has been modified to reflect that.

The PFT maps used in this study are available here :

https://orchidas.lsce.ipsl.fr/dev/lccci/orchidee_pfts.php

Table A2:    Is the partial derivative  averaged over space and time?  Therefore the 4dEnvar itself doesn't account for any seasonality (time-variation) in the relationship?

The partial derivative in Table A2 is indeed averaged in space and time; it is used for the e-4DVar approach to select the "best" epsilon values, It is not used with 4DEnVar. In both approaches, assimilation is performed over the entire two-year time window. The temporal variation is therefore taken into account. We have added this clarification to the text of Table A2 and Figure A2.

> **Spatial and temporal average** of the partial derivative for all parameter for each PFT

Figure 7: I was surprised at how well the True – Posterior 4dEnvar net carbon flux (top right panel) performed given the limitation of the spatial footprint influencing the station CO2, thus informing the biogenic contribution to CO2. This is promising, but be aware, that the ability to match the net carbon flux gives no guarantee that the component fluxes are well simulated. An interesting complement to this plot would be to compare the true component fluxes of GPP and ecosystem respiration against the posterior component fluxes of GPP and ecosystem respiration for the complex case.

We agree that the performance of True - Posterior 4dEnVar was significant on both net and gross (not shown in the first version of the paper) carbon fluxes. We have now included in the appendix the analysis on GPP (figure A4) which shows consistent quality of the fit of 4DEnVar posterior to the True observation.

We believe that this result is mainly due to the fact that we are in a twin experiment with no model-data bias. In this case, the model is considered 'perfect' and can fully recover the synthetic observation, which we believe greatly simplifies the problem. We also believe that the poorer performance of e-4DVar is mainly due to the fact that 1) the method is more sensitive to local minima, 2) the estimation of the tangent linear model relying on a finite difference approach is more uncertain (strongly dependent on the value of epsilon),and 3) The BoND PFT does not change because the PFT is not sufficiently sensitive and is therefore compensated by other PFTs/regions.

 However, both of these problems seem to be solved by 4DEnVar. The approach is less sensitive to local minima because it generates an ensemble that is less affected by local minima. We added the figure of the GPP estimates in the appendix (Figure A3) and added this text in discussion L491

> **Fig. A3 shows the differences in spatial distribution of gross primary production (GPP) between the "synthetic" fluxes and the prior/posterior estimate of the two methods, as well as their global yearly budget. We can see that GPP obtained with the 4DEnVar method is slightly better than the e-4DVar method for the global budget and better matches the spatial distribution of the synthetic flux.The e-4DVar method appears to compensate for the lack of change between the prior and posterior GPP across most of the Northern Hemisphere.**

and L497

This experiment of calibrating a large number of parameters represents a more realistic case, even if we consider **a very low** model/observation error. **The results demonstrate the good performance of 4DEnVar, which, even in a 'perfect' model situation, i.e. a model that can perfectly simulate observations, can assimilate observations while being less impacted by local minima. However, this may not be the case when using actual observations and introducing more complex modelling/observation errors.**

Line 431: I think a more nuanced discussion of equifinality is required here and/or in the Discussion. In addition to the pure number of parameters attempted to calibrate simultaneously – equifinality can arise for a number of different reasons – 1) a single parameter type being compensated within the large list of PFTs, 2) the station $CO_2$ concentration is influenced through the NBP, which is a confluence of both photosynthetic and respiration processes, which can easily compensate for each other to provide a net biogenic carbon flux consistent with station $CO_2$ data. I understand that this is a twin model experiment, so the following do not necessarily contribute here, but if this setup were to be applied to real data additional equifinality challenges present themselves including 1) the model state itself (carbon, water nutrient pools) have not been constrained by any data, thus parameters will compensate for biases due to model state initialization problems 2) The biogenic fluxes (controlled by parameters) would seem to contribute just a portion of the land-atmosphere carbon exchange which includes other large fluxes from fossil fuel, fires and ocean fluxes which would have to be measured accurately-- 3) atmospheric model transport errors, influencing the relationship between land carbon flux and station $CO_2$ data.

We agree with the reviewer and propose adding the following text:

L507

**In this twin experiment, both methods have to deal with the inherent equifinality of atmospheric concentration assimilation. This equifinality occurs when parameters compensate for each other, resulting in either an incorrect spatial distribution of NBP or inaccurate estimates of subcomponents such as GPP and total ecosystem respiration (TER), but**

still allowing for a match with observations. Although both methods considered in this study successfully recovered the global budgets for NBP and GPP, the e-4DVar method did not obtain the correct spatial distributions of NBP and GPP (see Figures 7 and 8). This is not the case for the 4DEnVar method, which better recovered the 'true' spatial distributions of NBP and GPP. We believe that this equifinality could increase the number of local minima, further disrupting the performance of the e-4DVar method. We also believe that the ensemble nature of the 4DEnVar method provides a more comprehensive view of the parameter space, making it less sensitive to local minima and therefore to equifinality issues.

L553

The assimilation of real observations of atmospheric concentrations may also increase the equifinality mentioned in Section 4.1 for several reasons, such as: i) Incorrect initial conditions of the carbon pools, which can impact respiration; ii) Wrong estimates of other flux components, such as ocean or fossil fuel components; iii) Structural errors in either the land surface model or the transport model. The issue of incorrect initial conditions can be addressed by starting the simulation a couple of years before the assimilation window. This allows for the correction of the initial carbon pool and better accounts for the effects of the new parameter values on the carbon pool. To handle other components, such as ocean components, the same assimilation can be repeated using different estimates of the ocean flux. Ideally, an ocean model could be included in the optimization to calibrate both land and ocean components, as is done in atmospheric inversion. The advantage of the 4DEnVar method is that it only requires forward simulations. Therefore, no code adaptations are needed, making it easier to use different transport models. This should help detect and address structural errors. The equifinality can also be reduced by assimilating multiple data streams simultaneously, as done in Peylin et al. (2016) and Bacour et al. (2023), to calibrate both GPP and NBP at the same time.

Line 512:  Given the significant challenges related to equifinality mentioned above, I am not sure this setup shows "great potential" to constrain parameters.   I might be more realistic and state that it demonstrates that 4DEnvar shows more potential than eta-4DVar.

We continue to believe that the 4DEnVar method has strong potential. Mainly because the technical implementation of this is simpler than that of the standard 4DVar method and the size of the ensemble required is reasonable, with the methods giving satisfactory results (good reduction of the model observation mismatch). We believe that many of the reviewer's very relevant comments are inherent to atmospheric concentration assimilation and would therefore still be present with other assimilation methods. We mostly agree with the reviewer and have replaced "**great**" with **'good'** to be more moderate.

Purely technical corrections

Abstract:

Awkward:  "These models rely on parameterisations that necessitate to be carefully calibrated".   These models rely on parameterizations that require careful calibration.

Thank you for the suggestion; We have applied this correction.

Introduction:

Can you describe in terms more accessible to the general community what isotropic means in this context?

"corrections to CO2 surface fluxes are isotropic in time and space."

We added in the text L32

> This statistical optimisation generally assumes that the corrections to CO2 surface fluxes are isotropic in time and space. **This suggests that errors in surface fluxes are only correlated in space by distance between points, and not by direction. Furthermore, these errors are not strongly correlated in time.**

Line 115:   I wouldn't use the terminology 'assimilation' routine to describe photosynthesis or carbon uptake routine.  Assimilation is often used within data assimilation context, a component of this analysis, which is not what this is describing.

We agree with the reviewer and have changed 'The carbon assimilation' by 'The carbon fixation'.

Section 2.1.4.  I think it's also worth mentioning that you didn't optimize the prior biogenic fluxes by constraining them with carbon pool observations (LAI, biomass, soil carbon etc).

We presented this aspect in section 2.1.1 describing the spin up, transient and historical simulation carried out to equilibrate the soil carbon pool that are computed dynamically in ORCHIDEE, which also equilibrate other carbon pools such as the leaf area index (LAI) and biomass. We believe that, thanks to the other changes requested by the reviewer, we have presented these aspects more clearly.

Shouldn't Figure A1 include the land biogenic fluxes for the truth simulation, just for relative perspective?  After all the parameter optimization is based on the influence of biogenic fluxes on the atmospheric CO2.

This figure is intended to show the other components of the global C budget that are used as input to LMDZ (described in §2.1.4) that have not been optimised in our study. We present the actual simulation of the net land biogenic fluxes in Figure 7. We believe that including these elements in Figure A1 could cause confusion as to what has been optimised in our framework and what has not. We therefore prefer not to include terrestrial biogenic fluxes in Figure A1.

Line 305:  LAImax:   The absolute max value that LAI can be? Can you clarify?  Does this mean carbon cannot be allocated to leaf carbon once achieving this level?

LAImax is the maximum value that LAI can reach for each PFT, which stops allocation to the leaves once this value is reached. We added this clarification L323
**Once the LAI reaches LAImax, no carbon is allocated in the leaf.**

Line 444:  LAI or LAImax ?

We have corrected the manuscript to add LAImax.

---

## Author Comment (AC2)

**Review #3**

Summary of manuscript

The authors explore the assimilation of atmospheric $CO_2$ concentration data for parameter calibration of the ORCHIDEE model using the 4DEnVar data assimilation method. Through carefully designed data assimilation experiments, they demonstrate the capability of 4DEnVar in assimilating atmospheric $CO_2$ concentration for parameter calibration, and highlighted its superiority over the ε-4DVar method in terms of computational efficiency, parameter recovery, and fitting to $CO_2$ concentration observations.

We would like to thank the reviewer for proofreading the manuscript, for their insightful comments that helped improve the article, and for pointing out numerous typos. (Please note that changes to the text are referenced by the line number in the new manuscript.)

General comments

Due to the continuous evolution of land surface models (LSMs), the use of 4DVar and gradient descent method for calibrating LSM parameters faces significant challenges in maintaining the tangent linear and adjoint models. Thus, it is necessary to explore adjoint-free variational methods. However, as the authors mentioned, the results obtained here for the ε-4DVar are not equivalent to a standard 4DVar, and no conclusions can be drawn regarding the comparison between the 4DEnVar and standard 4DVar methods. In light of this, to what extent can this study provide insights and practical guidance for the application of 4DEnVar to other LSMs and the assimilation of real, multi-source observations? I believe the manuscript would benefit from a clearer articulation of its research significance.

We thank the reviewer for this comment and agree that better stating the

significance of the research will help the paper have a strong impact. While we do not directly compare the results to 4DVar, we do compare to ε-4DVar, which has been used as a surrogate for 4DVar in our community due to the difficulty in maintaining the tangent linear/adjoint of the model. We show in this study that 4DEnVar can be used to assimilate atmospheric CO2 data, and we compare the ε-4DVar method in order to strengthen this message. We agree that this message may be missed in the article and propose the following addition text L 534:

> **The assimilation of atmospheric CO2 concentration data using 4DVar has been implemented with a tangent linear model, as in Castro-Morales et al., 2019, or an adjoint model, as in Scholze et al., 2007. In these cases, the tangent linear or adjoint model was developed alongside the forward model. However, the ε-4DVar method was used in experiments where obtaining the tangent linear or adjoint model proved too difficult, such as in Peylin et al., 2016, and Bacour et al., 2023. Although ε-4DVar is not equivalent to standard 4DVar, a comparison of 4DEnVar with ε-4DVar demonstrates the strong performance of 4DEnVar, making it a promising candidate for this application.**

We also modify the following paragraph L580 by:

> **Moreover, the 4DEnVar method was used to assimilate several types of data using either simple carbon model (Douglas et al., 2025) or more complex LSM as the JULES LSM (Pinnington et al., 2020, 2021; Cooper et al., 2021). This new application in the ORCHIDEE LSM shows that this method is model-independent. By adding different observation terms (one term per data flux) to the cost function, the method should be able to perform multi-flux data assimilation, which would help to reduce the equifinality**

**problem.**

The manuscript focuses on the introduction, application, and evaluation of the 4DEnVar and ε-4DVar methods throughout the methodology, results, and discussion sections. However, this focus is not well reflected in the title. Perhaps the authors could consider revising the title in light of related works, such as Yaremchuk et al. (2016).

We thank the reviewer for this comment, however, we believe the title does accurately reflect the content and outcome of our paper. This work was conducted to prepare for the assimilation of atmospheric $CO_2$ concentration data in a Land Surface Model (LSM), which offers numerous advantages as presented in the introduction. However, this type of assimilation is not straightforward and requires the development of a robust data assimilation (DA) system. Our article focuses specifically on this DA system, which is why the title begins with 'Toward the.' While we do not present actual assimilation of atmospheric $CO_2$ concentration data, we propose promising methodologies for future application. We particularly emphasize the use of Adjoint-free Variational Methods, as many complex LSMs like ORCHIDEE, JULES, CLM cannot rely on adjoint models. We believe it is important for the community to know that this type of assimilation is possible with a system that is easier to implement and requires only forward simulations. The comparison with the ε-4DVar method was included because it has been previously used in ORCHIDEE (Peylin et al., 2016; Bacour et al., 2023), making it a relevant benchmark to strengthen our message. Unlike the significant work of Yaremchuk et al. (2016), our study focuses exclusively on Adjoint-free Variational Methods (as ε-4DVar does not require an adjoint).

In the comparison of the assimilation results between the 4DEnVar and ε-4DVar methods, the authors repeatedly attribute the poorer performance of the ε-4DVar method to the fact that it falls into a local minimum. However, for the 4DVar method, whether the parameter iteration converges to a local minimum undoubtedly depends on factors such as the a priori parameter vector. This study employed only one a priori parameter vector, and its generation process was not clarified. This raises concerns about the reproducibility and generalizability of the findings. In other words, would different conclusions be reached if a different a priori parameter vector was used?

No other prior was used as the kind of assimilation are expensive therefore only one optimisation is performed (see for example Peylin et al., 2016; Schürmann et al., 2016; Castro-Morales et al., 2019; Bacour et al., 2023). We clarify how the prior was generated L296:

> **A new vector of a priori parameters was generated manually, ensuring that it differed from the "real" parameter values while retaining physically meaningful values.**

However, we also note that the poorer performance of the e-4DVar is also due to certain regions that are less well monitored by atmospheric stations in L486, as shown in Figure 7 and A3.

> **We believe that the different spatial structure obtained by ε-4DVar is likely to be explained by the fact that the two PFTs TrBE and BoND are not well monitored, creating a dipole in the Amazonian and Siberian regions to compensate for the erroneous carbon flux in other regions.**

A more detailed description and presentation of the methods and results are needed. The manuscript currently lacks an explanation of the parameter set's

value range and sampling approach. It would be beneficial to include formulas that demonstrate how the selected parameters influence ecosystem processes such as photosynthesis, respiration, and other carbon cycle components. Personally, I would appreciate seeing the distribution of the parameter ensemble and the spread of the ensemble simulations, as presented in Pinnington et al. (2020).

We completely agree with the reviewer that showing the prior and posterior parameter uncertainties is a vital part of this type of work, and indeed is a key strength of the 4DEnVar method. These aspects have also been noted by other reviewers, and we apologise for omitting the part of the methods used to generate the sample. Figures 3 and 6 have been modified to show the standard deviation of the *a priori* and *a posteriori* distributions for the 4DEnVar methods, as well as how the ensemble were generated in accordance with the work of Douglas et al. 2025. And added the following text and in L463:

> **Furthermore, Fig. 3 shows a significant decrease in the standard deviation of the posterior ensemble. This allows us to identify which parameters and therefore which PFTs appear more sensitive. In this case, it seems that the results for the TrNC3 and Crops C4 PFTs are the most uncertain.**

and in L473:

> **The posterior ensemble generated for the 4DEnVar also shows a reduction in uncertainty for all parameters. This uncertainty reduction is not equal for all parameters - a maximum reduction can be seen for the Q10 parameter(reducing the standard deviation by 94 %) and the lowest for the less sensitive $m_{maint.resp}$ parameter (with a 14% reduction for the NC4 PFT).**

We have chosen not to include the equations of the model involving the parameters to be optimised for the sake of clarity, especially given the complexity of the ORCHIDEE model. We have shared all previous work

describing the model in detail in section 2.1.1. We have also explained the choice of calibrated parameters and their respective relationship with the main processes in which they participate, as detailed in sections 2.3.1 and 2.3.2. The impact of each parameter on atmospheric CO2 concentration is examined in section 2.4. We believe that describing the many model equations would not really improve the main messages of the article.

The authors may need to consider citing and discussing some recent studies, such as Douglas et al. (2025).

The paper of Douglas et al. (2025) was discussed in L86

**This method was also successfully used by Douglas et al. 2025 to calibrate the parameters of a simple carbon model in a twin experiment.**

in L274:

**A posterior ensemble can be obtained as it is described by Douglas et al 2025 by calculating Xa' where**

$$X'_a = X'_b (I + (HX'_b)^T R^{-1} HX'_b)^{-\frac{1}{2}}$$

Specific comments

Line 34: "i.e." to "i.e.,".

Thank you for spotting this error. We corrected the text accordingly.

Lines 50-51: Pay attention to the spacing before or after the paragraph.

We corrected the text following the reviewer's suggestion.

Line 54: "4DVar" to "four-dimensional variational (4DVar)". Please check the

use of abbreviations in the manuscript to ensure they are correct.

We have included this to the manuscript.

Lines 56-58: It is necessary to add references here, such as Talagrand and Courtier (1987).

We agree with the reviewer and added the references

Line 83: The citation format is incorrect and needs to be changed from "Pinnington et al. (2020)" to "(Pinnington et al., 2020)".

We have corrected the manuscript, thank you for spotting this.

Line 92: The space between 'approaches' and ',' is extra.

We have corrected the manuscript, thank you for spotting this.

Line 92-94: The sentence is not concise and clear. It is recommended to revise it as follows: "Although tangent linear or adjoint models are not required for methods such as GA, MCMC, or emulator-based approaches, these methods necessitate defining a large ensemble, making them unfeasible for use in this study due to the time-consuming nature of model simulations."

We agree with the reviewer. We revised the text to include this modification L92

Line 123: The period currently at the beginning of the line should be placed at the end of the previous line.

We have edited the manuscript to correct all typographical errors and

rephrase certain sentences.

Line 151 and Figure 1: You mentioned the stations are selected according to their 6-month averaged sensitivity. Which six months were chosen? Given seasonal variations, it would seem more reasonable to select a full year or multiple years. Additionally, you may need to clarify whether any climate pattern, such as ENSO or IOD, occurred during the sensitivity analysis period and the simulation period. Please provide a more detailed description.

We acknowledge that the station selection could have been explained in more detail. First we selected continental stations operating during the 2000-2001 period. For those stations, we evaluated the daily average sensitivity over the last six months for each month of the 2 year assimilation window. Then, we calculated the average sensitivity over the last six months (which corresponds to an average of 24 maps, each map representing the average sensitivity over the last six months). We have added in the legends of the Fig.1

> **Monthly mean sensitivity map of atmospheric CO2 concentrations to land carbon fluxes at the 21 stations considered over the 2000-2001 period. The average sensitivity map is obtained by deriving, for each atmospheric station and each of the 24 months, the map of the average daily sensitivity of the atmospheric concentration of CO2 to surface carbon flux (in ppm/GtC) over the last six months, and then calculating the average of the 24 maps.**

To our knowledge no ENSO or IOD events occurred during this period.

Line 160: The version of the Global Fire Emission Database used in the study is outdated, or why used this one?

We acknowledge that this dataset is not up-to-date. However, because this study relies on simulated data only, we believe that using a more recent

version of the database would not affect the study's conclusion or it's key message . However, we have been working on an updated configuration of the data assimilation framework aiming to assimilate real data. In doing so, we are considering updated datasets for the component of the surface $CO_2$ fluxes other than the biospheric one (including GFED) as well as an updated version of the atmospheric transport model.

Line 171: Please verify that the equations are correctly written. For example, vectors should be in italics, while matrices should not.

Thank you for spotting this. We have corrected all incorrect notations.

Line 278: It is suggested to consider organizing the default parameter values in a table and placing them in the supplement.

We have added a table to the appendix in order to show the True and Prior parameter values.

> We simulate the NBP fluxes at the global scale using the ORCHIDEE LSM with the default parameter values **(see Tab. A2)**

Line 284: It is necessary to specify how the a priori parameter vector was obtained.

The new a priori parameter vector was manually perturbed in order to find a new set of parameters giving a simulation of the a priori atmospheric concentration consistent with the observations.. We added to the text L296:

> **A new vector of a priori parameters was generated manually, ensuring that it differed from the "real" parameter values while retaining physically meaningful values.**

Lines 289-290: "Vcmax" and "°C" should not be italicized.

We have corrected all incorrect notations in italics in the manuscript.

Line 312: It is recommended to provide some references regarding this setup.

Lines 313-314: A more detailed explanation of the parameter range settings and the rationale behind them is needed.

We added to the text some references and justification in L334.

**The configuration of the R and B matrices was based on previous data assimilation studies with ORCHIDEE and a simplified carbon model (Kuppel et al., 2012 and 2013; MacBean et al., 2016; Peylin et al., 2016; Bastrikov et al., 2018). These studies employed diagonal matrices for R and B to assimilate in situ observations, while Peylin et al. (2016) specifically used them for atmospheric $CO_2$ observations.**

Lines 364-365: Although RMSD and MAD are common statistical metrics, I still recommend that the authors provide their calculation formulas and explanations here. Since the observations have already been synthesized, are the simulation results involved in the calculation also synthesized? Furthermore, both RMSD and MAD, in terms of their form, resemble the cost function, as they include the critical term representing the difference between observations and simulations. In assimilation experiments, reductions in these metrics are expected. It would be valuable to explore additional metrics with distinct physical interpretations (e.g., coefficient of determination, $R^2$) to comprehensively assess method performance.

We added an Appendix named **Metrics calculation**

**The RMSD and MAD are calculated as follows:**

$$RMSD = \sqrt{\frac{1}{N_t} \sum_{t=0}^{N_t} \left(\mathcal{H}(\boldsymbol{x_*})_t - \boldsymbol{y}_t\right)^2},$$

$$MAD = \frac{1}{n_{param}} \sum_{i=0}^{n_{param}} |\boldsymbol{x_{*i}} - \boldsymbol{x_{true i}}|,$$

**where x\* can be either xb or xa. The Pearson correlation coefficients were computed using the Numpy Python library with the 'corrcoef' function. The paired t-tests were computed using the 'stats.ttest\_rel' function from the Scipy library.**

We agree with the reviewer that a range of metrics is necessary to evaluate performance. This was also requested by the second reviewer.

In addition to the evaluation of the NBP already in the manuscript, we computed to Pearson correlation coefficients of the NBP in time and in space and add this to the manuscript in L 442:

**The Pearson correlation coefficient between the 'synthetique' NBP and the prior NBP is 0.87 in time and 0.17 in space. The posterior NBP obtained by the 4DEnVar method shows a Pearson correlation coefficient against the 'synthetique' NBP of 0.99 in time and 0.98 in space. In comparison, the posterior NBP obtained by the e-4DVar method has correlation coefficients of 0.98 in time and 0.84 in space.**

We added the figure of the GPP estimates in the appendix (Figure A3) and added this text in discussion L491

**Fig. A3 shows the differences in spatial distribution of gross primary production (GPP) between the "synthetic" fluxes and the prior/posterior estimate of the two methods, as well as their global yearly budget. We can see that GPP obtained with the 4DEnVar method is slightly better than the e-4DVar method for**

**the global budget and better matches the spatial distribution of the synthetic flux.The e-4DVar method appears to compensate for the lack of change between the prior and posterior GPP across most of the Northern Hemisphere.**

We have also added Figure A4, which shows the time series for the different stations. We calculated Pearson's correlation coefficients for each station, but they were all between 0.98 and 1. This high value is mainly due to the fact that we are in a twin experiment mode, where 'synthetic' observations are generated by the model. We felt that it was not necessary to include them in the manuscript.

Line 367: There should be a space between the number and the unit.

We have corrected the manuscript.

Line 370-372: I don't fully understand why configurations with more ensemble members (e.g., 350, 400) result in a smaller RMSD reduction. Could the authors provide an explanation?

We thank the reviewer for this very interesting question, which is not yet fully understood. We have added some hypotheses that we have formulated to the text L519.

**But the performance of the 4DEnVar method seems dependent on the generated ensemble. As shown in Table 2, slightly lower performance is observed with larger ensembles, indicating that a bigger ensemble does not necessarily yield better results. This could be due to the increased dimensionality of the problem, making the iterative minimization more challenging.**
**Additionally, we generated a new ensemble for each experiment, which provides different information about the parameter space**

**and can lead to different optimal values. This shows the importance of the prior ensemble generated. Nevertheless, the reduction in RMSD remains satisfactory, with a reduction of more than 90%.**

Line 395: The use of 'seem to' here makes the experiment appear insufficiently rigorous.

We agree and have removed 'seem to'.

Line 412: Use exponential notation and change "GtC/year" to "Gt C year$^{-1}$".

All notations (including those in the figure) have been modified.

Line 523: The line break in the link seems to be problematic.

We thank the reviewer and have corrected the manuscript.

Figure 1. The website should include the date of the last access.

We have corrected the manuscript.

Figure 3. "triangle" to "triangles".

We have corrected the manuscript.

Figure 4: It is recommended to consistently retain two decimal places.

We thank the reviewers for these comments. We have modified Figure 4 to show only two decimal places.

Figure 7: The presentation of the last subplot can be improved, for example, the current color scheme does not match well with that of the other subplots.

We thank the reviewer for this comment

Figure A1: Revise unnecessary capitalization and add a comma at the end of the sentence.

We have corrected the manuscript.

References

Douglas, N., Quaife, T., and Bannister, R.: Exploring a hybrid ensemble–variational data assimilation technique (4DEnVar) with a simple ecosystem carbon model, Environmental Modelling & Software, 186, 106361, https://doi.org/10.1016/j.envsoft.2025.106361, 2025.

Pinnington, E., Quaife, T., Lawless, A., Williams, K., Arkebauer, T., and Scoby, D.: The Land Variational Ensemble Data Assimilation Framework: LAVENDAR v1.0.0, Geosci. Model Dev., 13, 55-69, 10.5194/gmd-13-55-2020, 2020.

Talagrand, O. and Courtier, P.: Variational Assimilation of Meteorological Observations With the Adjoint Vorticity Equation. I: Theory, Quarterly Journal of the Royal Meteorological Society, 113, 1311-1328, https://doi.org/10.1002/qj.49711347812, 1987.

Yaremchuk, M., Martin, P., Koch, A., and Beattie, C.: Comparison of the adjoint and adjoint-free 4dVar assimilation of the hydrographic and velocity observations in the Adriatic Sea, Ocean Modelling, 97, 129-140, https://doi.org/10.1016/j.ocemod.2015.10.010, 2016.

Bastrikov, V., Macbean, N., Bacour, C., Santaren, D., Kuppel, S., and Peylin, P.: Land surface model parameter optimisation using in situ flux data: Comparison of gradient-based versus random search algorithms (a case study using ORCHIDEE v1.9.5.2), Geoscientific Model Development, 11,

https://doi.org/10.5194/gmd-11-4739-2018, 2018.

Peylin, P., Bacour, C., MacBean, N., Leonard, S., Rayner, P., Kuppel, S., Koffi, E., Kane, A., Maignan, F., Chevallier, F., Ciais, P., and Prunet, P.: A new stepwise carbon cycle data assimilation system using multiple data streams to constrain the simulated land surface carbon cycle, Geoscientific Model Development, 9, https://doi.org/10.5194/gmd-9-3321-2016, 2016.

Castro-Morales, K., Schürmann, G., Köstler, C., Rödenbeck, C., Heimann, M., and Zaehle, S.: Three decades of simulated global terrestrial carbon fluxes from a data assimilation system confronted with different periods of obse;rvations, Biogeosciences, 16, https://doi.org/10.5194/bg-16-3009-2019, 2019.

Kuppel, S., Chevallier, F., and Peylin, P.: Quantifying the model structural error in carbon cycle data assimilation systems, Geoscientific Model Development, 6, https://doi.org/10.5194/gmd-6-45-2013, 2013.

MacBean, N., Peylin, P., Chevallier, F., Scholze, M., and Schürmann, G.: Consistent assimilation of multiple data streams in a carbon cycle data assimilation system, Geoscientific Model Development, 9, https://doi.org/10.5194/gmd-9-3569-2016, 2016.

Bacour, C., Macbean, N., Chevallier, F., Léonard, S., Koffi, E. N., and Peylin, P.: Assimilation of multiple datasets results in large differences in regional-to global-scale NEE and GPP budgets simulated by a terrestrial biosphere model, Biogeosciences, 20, https://doi.org/10.5194/bg-20-1089-2023, 2023.

Scholze, M., Kaminski, T., Rayner, P., Knorr, W., and Giering, R.: Propagating uncertainty through prognostic carbon cycle data assimilation system simulations, Journal of Geophysical Research Atmospheres, 112, https://doi.org/10.1029/2007JD008642, 2007.

---

## Author Comment (AC3)

**Review #2**

While this manuscript addresses a highly relevant and important topic—adjoint-free data assimilation for land surface model parameter estimation using atmospheric $CO_2$ concentrations, there is a fundamental conceptual misunderstanding regarding the data assimilation framework employed. The authors repeatedly describe their method as a "4DEnVar" approach. However, after careful review of the methodology and experimental design, it is clear that the implemented framework aligns more closely with a 3DEnVar method rather than a true 4DEnVar. I will provide more specific comments below detailing the evidence for this classification error and offering suggestions for how to appropriately revise the manuscript.

We would like to thank the reviewer for taking the time to read and comment on the manuscript. We acknowledge the concerns raised regarding the terminology used to describe our method. However, we believe there may have been a misunderstanding about the nature of our approach. To clarify, our method is indeed a 4DEnVar and not a 3DEnVar, as it involves time assimilation over a two-year window covering all data.

We realise that our initial explanation may not have been sufficiently clear, and we apologise for any confusion this may have caused. In response, we have provided detailed replies to each of the reviewers' comments below and have revised the manuscript accordingly to improve clarity.

(Please note that changes to the text are referenced by the line number in the new manuscript.)

1. The authors incorrectly label their method as "4DEnVar." In classical data assimilation terminology, the key distinction between 3D and 4D variational methods lies in the incorporation of assimilation windows. A three-dimensional variational (3DVar or 3DEnVar) method assimilates observations as a function of space only,

without explicitly considering the time evolution of the model states. In contrast, a four dimensional variational (4DVar or 4DEnVar) method introduces a temporal dimension  by defining an assimilation window, allowing the model to evolve dynamically and best  fit the observations distributed across time within that assimilation window. In this  manuscript, there is no explicit mention of an assimilation window, nor is there any  evidence that the model trajectory evolves and interacts with observations at multiple  times during a window. Instead, observations appear to be treated statically, consistent  with a 3DEnVar framework. Therefore, the method used in this study should be  accurately referred to as 3DEnVar, not 4DEnVar. The manuscript must be revised to  correct this misclassification throughout, including the title, abstract, methods, results,  and discussion sections.

We fully agree with the explanation given by the reviewer on the difference between the 3DVar and the 4DVar. However we would like to clarify that our approach does fall under the category of the 4DVar as we assimilate pseudo-observations both in space and time. Specifically, we are using the full 2 years of pseudo-observations within an assimilation window of 2 years. Furthermore, a temporal window is essential when calibrating parameters. Since parameters remain constant over time, we need to find the 'best' set of parameters that fits the observations over time. Assimilating a single time step would lead to different parameters for each time step. In order to clarify this point, we have added to L300

> **The assimilation of atmospheric $CO_2$ concentrations at the 21 stations is performed simultaneously using a two-year assimilation window in order to assimilate all observations and thus monitor variations in carbon fluxes in space and time, as shown in Fig. A4.**

Figure A4 has been added to illustrate the assimilated time series and clarify this point.
Given that we are using two years of observations and that temporal evolution is as

important as the spatial distribution of carbon fluxes, we need to incorporate time into the assimilation scheme and therefore rely on a 4DVar or 4DEnVar method.

2. The description at L56–58 defines general variational assimilation, not 4DVar specifically. 4DVar uniquely involves an assimilation window and time-evolving model trajectory. Please correct this definition.

We apologise for the lack of precision. We have therefore modified it L57 as follows:

> The 4DVar approach involves defining a cost function (which is usually based on a least-square criterion) that computes the difference between observations and model outputs **distributed in space and time** as well as a background term that accounts for prior knowledge of the parameters.

3. Data assimilation can generally be categorized into full-field assimilation and anomaly assimilation. Full-field assimilation adjusts the model state towards observed absolute values, while anomaly assimilation only incorporates the observed anomalies. The authors must clearly specify which approach is used. If full-field assimilation is applied, the impact on climatology should be explicitly evaluated and presented.

We thank the reviewer for raising this point. In our study, the model parameters are adjusted to better match the model output to the absolute values of the observations, rather than considering their anomalies. As such, the assimilation most closely match the definition of a full-field assimilation. This approach is consistent with the goal of our study, which is to evaluate the potential of using atmospheric $CO_2$ observations to

constrain land surface model parameters through 4DEnVar.

We would like to emphasise that this study is based on a controlled twin experiment, where the primary objective is to assess the method's ability to recover known "true" parameter values and to match synthetic observations. As such, we focus on demonstrating the feasibility and strengths of the approach itself, rather than evaluating its impact on climatological metrics in a real-world context. We agree that when assimilating real observations, it will be important to assess the impact on other aspects of the model outputs. However, since this study is methodological in nature and uses synthetic data, we believe that including such an evaluation would be beyond the scope and would not directly strengthen the main message.

4. The cost function shown in Equation (5) corresponds to the standard 3DVar formulation, as it lacks any temporal dimension or model trajectory integration. A true 4DVar cost function should involve the evolution of the model state over time:

Please revise both the equation and the method name accordingly.

While the reviewer is correct that the standard 3DVar formulation lacks a temporal dimension, here, y represents a vector of observations and H(x) the model output over the given window. Therefore, the assimilation over time is made implicit by this formulation.However, we recognise that there may be some confusion about this and have changed the manuscript L185.

> **Here, the model operator output H(x) and the observation y are defined in time and space. All observations are concatenated into a large vector of observations y, in order to represent all observations available in a given time window. The same operation is performed for the output of operator H(x).**

We have added the classic 4DVar Cost feature and explain our simplification in L200:

the 4DVar cost function:

$$J(\boldsymbol{x}) = \frac{1}{2}\Big(\sum_t^{N_t} \boldsymbol{\mathcal{H}}_t(\boldsymbol{x}_t) - \boldsymbol{y}_t\Big)^T \mathbf{R}_t^{-1}(\boldsymbol{\mathcal{H}}_t(\boldsymbol{x}_t) - \boldsymbol{y}_t) + \frac{1}{2}(\boldsymbol{x} - \boldsymbol{x_b})^T \mathbf{B}^{-1}(\boldsymbol{x} - \boldsymbol{x_b})$$

**where t refers to time steps 0,...,N_t. Since the parameter must be constant over time, we consider only a single time window that includes all observation vector y (in time and space). We therefore simplify to the compact form the initial 4DVar cost to the compact form:**

$$J(x) = \frac{1}{2}(\mathcal{H}(x) - y)^T \mathbf{R}^{-1}(\mathcal{H}(x) - y) + \frac{1}{2}(x - x_b)^T \mathbf{B}^{-1}(x - x_b).$$

5. Lines 310-315: The R matrix represents the observation error covariance (not both the model/observation error) and should account for spatial heterogeneity in observation uncertainty. Setting all diagonal elements to 0.01 ppm is overly simplistic. Even in a simple setup, R could be statistically computed based on the variance of the observation. Please consider a more realistic design or justify this simplification.

In our case, model structural errors include both the structural errors associated with the ORCHIDEE model for the computation of the net carbon fluxes and the transport model error. In this context the model errors are likely the dominant part of the R matrix, given that measurements of atmospheric CO2 are usually relatively precise. The R matrix represents errors linked to the comparison of y and H(x) excluding the model parametric error that is accounted for in the B matrix. Therefore, the error in the model structure and the observation operator are essentially taken into account here. so that R contains model and observation errors in this case, and B only parameter errors.

Furthermore, we agree that setting all diagonal elements to 0.01 ppm is simplistic, but since we are in a conceptual framework relying on twin experiments, we have considered the model to be perfect, allowing for a very low error to be used. We would also add that this observation error is close to the difference in atmospheric $CO_2$ data between observation and model simulations obtained after optimisation (especially for 4DEnVar), as illustrated in Figure 4. We have added the following to

the text L329:

**Indeed, since no error was included in the pseudo-observation and as we are in a Twin experiment, a simplistic R matrix was used.**

We also added to the text some references and justification in L334.

**The configuration of the R and B matrices was based on previous data assimilation studies with ORCHIDEE and a simplified carbon model (Kuppel et al., 2012 and 2013; MacBean et al., 2016; Peylin et al., 2016; Bastrikov et al., 2018). These studies employed diagonal matrices for R and B to assimilate in situ observations, while Peylin et al. (2016) specifically used them for atmospheric $CO_2$ observations.**

6. Lines 364-366: Please clearly define how RMSD is computed, and indicate whether systematic bias is removed prior to calculation.

we have clarified how we computed the RMSD in L386 :

The results in terms of 1) mean reduction in the mean square difference (RMSD) **calculated between the pseudo-observation and the simulation over the two years of the assimilation window for the 21 atmospheric stations**,

We also added an Appendix named **Metrics calculation**:

**The RMSD and MAD are calculated as follows:**

$$RMSD = \sqrt{\frac{1}{N_t} \sum_{t=0}^{N_t} \left(\mathcal{H}(\boldsymbol{x_*})_t - \boldsymbol{y_t}\right)^2},$$

$$MAD = \frac{1}{n_{param}} \sum_{i=0}^{n_{param}} |\boldsymbol{x_{*i}} - \boldsymbol{x_{true i}}|,$$

**where x\* can be either xb or xa. The Pearson correlation coefficients were computed using the Numpy Python library with the 'corrcoef' function. The paired t-tests were computed using the 'stats.ttest\_rel' function from the Scipy library.**

7. The evaluation relies almost entirely on RMSD and MAD. Please consider providing additional diagnostic metrics, such as correlation coefficients between posterior and true fields, or skill scores, to offer a more complete picture of assimilation performance.

We agree with the reviewer that a range of metrics is necessary to evaluate performance. In addition to the evaluation of the NBP already in the manuscript, we computed to Pearson correlation coefficients of the NBP in time and in space and add this to the manuscript in L 442:

**The Pearson correlation coefficient between the 'synthetic' NBP and the prior NBP is 0.87 in time and 0.17 in space. The posterior NBP obtained by the 4DEnVar method shows a Pearson correlation coefficient against the 'synthetic' NBP of 0.99 in time and 0.98 in space. In comparison, the posterior NBP obtained by the e-4DVar method has correlation coefficients of 0.98 in time and 0.84 in space.**

We added the figure of the GPP estimates in the appendix (Figure A3) and added this text in discussion L491

**Fig. A3 shows the differences in spatial distribution of gross primary production (GPP) between the "synthetic" fluxes and the prior/posterior estimate of the two methods, as well as their global yearly budget. We can see that GPP obtained with the 4DEnVar method is slightly better than the e-4DVar for the global budget and better matches the spatial distribution of the synthetic flux. The e-4DVar appears to compensate for the lack of**

**change between the prior and posterior GPP across most of the Northern Hemisphere.**

We have also added Figure A4, which shows the time series for the different stations. We calculated Pearson's correlation coefficients for each station, but they were all between 0.98 and 1. This high value is mainly due to the fact that we are in a twin experiment mode, where 'synthetic' observations are generated by the model. We felt that it was not necessary to include them in the manuscript.

8. The differences in RMSD reductions between different methods and configurations are discussed, but no statistical tests are provided. Please include simple significance tests (e.g., paired t-tests) to assess whether the differences in RMSD reductions are statistically meaningful across stations.

We thank the reviewer for this suggestion. We have performed a paired t-test as suggested by the reviewer, which confirms that the differences in RMSD reduction are significant. We added the following text in L423:

> **Since the posterior RMSDs obtained were close, we performed a paired t-test (Student, 1908) between the two posterior RMSDs to determine whether they were significantly different. We obtained a t-value of -2.125 between the posterior RMSDs obtained by 4DEnVar and e-4DVar, with a p-value of 0.046. This confirms that the average posterior RMSD obtained by 4DEnVar is significantly lower than the posterior RMSD obtained by e-4DVar, with a confidence level of 95%.**

9. The comparison between 4DEnVar and ε-4DVar is repeatedly emphasized, but ε

4DVar is not equivalent to standard 4DVar. Please emphasize earlier and more clearly that the ε-4DVar results are only a rough approximation and that conclusions should not be generalized to comparisons with a full 4DVar system.

We do agree that the ε-4DVar is not equivalent to standard 4DVar and we actually explicitly say it in the manuscript L530

> The results obtained here for the ε-4DVar are not equivalent to a standard 4Dvar using a tangent linear and adjoint model. Therefore, we can draw no conclusions on the comparison between the 4DEnvar and standard 4DVar methods as was highlighted in Liu et al. (2008)

In order to clarify we added in L229 the following sentence:

> **Due to this approximation, ε-4DVar is therefore not entirely equivalent to standard 4DVar.**

10. All evaluations are performed against the same synthetic dataset used for assimilation. For robustness, a portion of the synthetic observations should be withheld during assimilation and used for independent validation.

We agree with the reviewer that this approach is relevant as a sanity check for data assimilation, and in particular when "real" observations are used. Because we here use synthetic data over a limited time window (2-years) and a limited number of stations, we have rather chosen to perform a complementary evaluation with respect to NBP and the subcomponents carbon flux (GPP) in Fig 7 and Fig 3A. This evaluation is described in the discussion section, L491

> **Fig. 3A shows the differences in spatial distribution of gross primary production (GPP) as well as their global estimates. We can see that 4DEnVar better matches the global estimates and the spatial distribution of the synthetic flux. It seems that e-4DVar has to compensate for the fact that the BoND PFT does not change**

**between the prior and the posterior and seems to fall into a local minimum.**

References

Pinnington, E., Quaife, T., Lawless, A., Williams, K., Arkebauer, T., and Scoby, D.: The Land Variational Ensemble Data Assimilation Framework: LAVENDAR v1.0.0, Geoscientific Model Development, 13, https://doi.org/10.5194/gmd-13-55-2020, 2020.

Liu, C., Xiao, Q., and Wang, B.: An ensemble-based four-dimensional variational data assimilation scheme. Part I: Technical formulation and preliminary test, Monthly Weather Review, 136, https://doi.org/10.1175/2008MWR2312.1, 2008.

Bastrikov, V., Macbean, N., Bacour, C., Santaren, D., Kuppel, S., and Peylin, P.: Land surface model parameter optimisation using in situ flux data: Comparison of gradient-based versus random search algorithms (a case study using ORCHIDEE v1.9.5.2), Geoscientific Model Development, 11, https://doi.org/10.5194/gmd-11-4739-2018, 2018.

Peylin, P., Bacour, C., MacBean, N., Leonard, S., Rayner, P., Kuppel, S., Koffi, E., Kane, A., Maignan, F., Chevallier, F., Ciais, P., and Prunet, P.: A new stepwise carbon cycle data assimilation system using multiple data streams to constrain the simulated land surface carbon cycle, Geoscientific Model Development, 9, https://doi.org/10.5194/gmd-9-3321-2016, 2016.

Kuppel, S., Chevallier, F., and Peylin, P.: Quantifying the model structural error in carbon cycle data assimilation systems, Geoscientific Model Development, 6, https://doi.org/10.5194/gmd-6-45-2013, 2013.

MacBean, N., Peylin, P., Chevallier, F., Scholze, M., and Schürmann, G.: Consistent assimilation of multiple data streams in a carbon cycle data

assimilation system, Geoscientific Model Development, 9, https://doi.org/10.5194/gmd-9-3569-2016, 2016.

---

## Author Response (AR2)

**Report review #2**

This manuscript presents an ensemble-based variational data assimilation framework aimed at calibrating parameters in the ORCHIDEE Land Surface Model using atmospheric CO2 concentration data. The authors refer to their method as a 4DEnVar approach and demonstrate its performance through synthetic twin experiments. However, after carefully reviewing both the manuscript and the authors' response to my previous comments, I remain increasingly convinced that the methodology implemented in this study more accurately reflects a 3DEnVar formulation rather than a true 4DEnVar. I strongly encourage the authors to revisit my earlier review, in which I raised key concerns regarding the temporal treatment of the observation operator and the background error covariance. In the following comment, I will explain in more detail why the method, as described, lacks the defining features of a four-dimensional ensemble-variational system and should therefore be categorized as a 3DEnVar.

We would like to thank the reviewer for taking the time to read through and comment on this manuscript. Your comments will undoubtedly strengthen the paper and help clarify key points. We have thoroughly revised the manuscript and incorporated all the changes you suggested. Below, you will find a point-by-point response to your comments:

1. In my previous review, I highlighted that one of the fundamental distinctions between 3DVar and 4DVar (and, by extension, 3DEnVar and 4DEnVar) lies in the incorporation of an assimilation window, i.e., whether the assimilation framework explicitly accounts for the temporal evolution of observations and background states within a defined time window. However, in the initial manuscript, there was no mention or implementation of an assimilation window, and the cost function used follows a 3DVar formulation. In the authors' response, they argue that their assimilation window spans two years (2000–2001), equating the entire simulation period with a single "assimilation window". I must emphasize that this interpretation is incorrect and conceptually flawed. An assimilation window refers to the temporal period over which observations and model background states are compared and assimilated within each cycle. In typical 4DVar or 4DEnVar systems, the assimilation window is short (e.g., daily or sub-daily), and the system progresses through multiple assimilation cycles to iteratively improve the estimate of the

optimal analysis over time. In contrast, the authors perform only a single assimilation cycle over a two-year period, without any time-evolving ensemble perturbations or temporally resolved assimilation updates. If one were to accept the authors' definition of an assimilation window, then any 3DVar system operating on a long time series would be mistakenly classified as a 4DVar system, which is clearly not consistent with established data assimilation literature. Moreover, given that the model is run only once over 2000–2001, the proper assimilation window—had this been a genuine 4DVar or 4DEnVar system—should be of at least daily resolution within that two-year period, requiring many sequential assimilation cycles. Therefore, I cannot accept the authors' interpretation that their assimilation window is two years in length. Based on the methodology described and implemented, I am confident that this system is best characterized as a 3DEnVar, not a 4DEnVar.

Thank you for your thorough and insightful feedback regarding the classification of our data assimilation methodology. We greatly appreciate your expertise and the opportunity to refine our manuscript to ensure clarity and alignment with established data assimilation terminology.

We understand your concern that our interpretation of a 4DVar does not correspond to the conventional definition of a 4DVar (or 4DEnVar) system, which typically involves several short assimilation cycles (e.g., daily or sub-daily) in order to account for the temporal evolution of observations and model states. Your comment that an assimilation window refers to the time period during which observations and model background states are compared within each cycle is relevant, particularly in the context of traditional 4DVar systems used in meteorological applications.

In our study, we seek to optimize time-invariant parameters (e.g., Vcmax, Q10) using a variational framework that explicitly takes into account the temporal evolution of observations (atmospheric CO2 concentrations) and model results (CO2 concentrations simulated from the ORCHIDEE model coupled with the LMDZ atmospheric transport model) over a two-year period. These time-invariant parameters are fixed constants used to calculate the interested variable (here the NBP flux, and then the atmospheric concentration). You are therefore correct in saying that the background terms are not evaluated over time (as this is not possible), even if changing the parameter would alter the evolution of the NBP (and then the atmospheric concentration) over time (as this would change the state of the carbon pool, for example).

We also acknowledge that our use of a single assimilation cycle covering the entire two-year period deviates from the conventional 4DVar/4DEnVar framework, which typically uses sequential cycles with shorter assimilation windows.

However, the cost function incorporates observations spread over time and model forecasts, comparing them at several points during this period in order to constrain the parameters. It is this temporal dimension of the assimilation process that led us to classify our approach as 4DVar and 4DEnVar, as it captures the dynamic evolution of the system rather than relying on a single snapshot, as in 3DVar or 3DEnVar.

To avoid potential misinterpretation and to align more closely with the expectations of the data assimilation community, we propose a compromise by adopting the terms Variational Data Assimilation (VarDA) and Ensemble Variational Data Assimilation (EnVarDA) in place of 4DVar and 4DEnVar, respectively. These terms emphasize the variational nature of our approach, which optimizes parameters by leveraging the temporal evolution of observations and model outputs. We have revised the manuscript to reflect this updated terminology and have added a clarification in Section 1 to explicitly describe our assimilation framework L52:

There is a long history of using data assimilation frameworks to calibrate LSM parameters (Rayner, 2010; MacBean et al., 2022; Raoult et al., 2024b). **Most of the methods used for parameter** calibration are derived from Bayesian formulations of inverse problems and defined here as variational data assimilation (VarDA) methods. The VarDA method is inspired by the four-dimensional variational (4DVar) method, which was originally developed in the field of meteorology and Earth sciences (Talagrand and Courtier, 1987; Courtier et al., 1994; Asch et al., 2016) and also employed in atmospheric inversions to correct the surface CO2 fluxes (Chevallier et al., 2005; Basu et al., 2013; Liu et al., 2021). This approach is characterised by the definition of a cost function, which is typically based on a least-squares criterion. This cost function calculates two terms: (i) an observation term that computes the difference between observations and model outputs, and (ii) a background term that incorporates prior knowledge of

the state. The computation of both terms is performed in space and time. We define here the VarDA method, as our focus is not on directly optimizing the prior state. Instead, we concentrate on time-invariant parameters used in the parameterisation that defines the variable of interest, such as the Net Carbon Flux. Therefore, while the observation term of the cost function incorporates time-distributed observations and model predictions comparing them across multiple time points - the background term only compares prior parameter values once, as these values remain constant over time. Furthermore, with the VarDA method, a single assimilation cycle covering the entire observation period is used, which differs from the conventional 4DVar framework, which generally uses sequential cycles with shorter assimilation windows. In order to minimise this cost function, the VarDA method calculates its gradient with respect to the different parameters to be calibrated.

and L89

More recently, an ensemble 4DVar method named 4DEnVar implemented in Pinnington et al. (2020) for LSM parameter estimation has proved very promising. This method uses a small ensemble to circumvent the necessity for a tangent linear and adjoint model. This 4DEnVar method has been used to estimate JULES LSM crop parameters at a single Nebraskan site (Pinnington et al., 2020) and to calibrate pedotransfer functions to improve JULES LSM soil moisture predictions over East Anglia (Pinnington et al., 2021) and the whole of the UK (Cooper et al., 2021). This method was also successfully used by Douglas et al. (2025) to calibrate the parameters of a simple carbon model in a twin experiment. Although the method was defined as 4DEnVar in Pinnington et al. (2020) and Douglas et al. (2025), we choose to refer to it as EnVarDA to maintain consistency with the definitions previously presented.

This clarification highlights that our approach optimizes parameters over a time period using a single cycle, with the cost function incorporating time-varying observations and model outputs, distinguishing it from traditional 3DVar/3DEnVar methods while acknowledging its differences from standard 4DVar/4DEnVar implementations.

We have also added:

L205: "where, for example, the concatenated vector  $y=(y_0,y_1,...,y_{N_t})^T$  represents all available observations at all times over the time window." L261: "where each  $H(x_i)$  is a concatenated vector of extracted simulations to correspond with all observations available at all times across the time window.

We believe this revision addresses your concerns while maintaining the integrity of our methodology. Thank you again for your valuable feedback, which has significantly improved the clarity and rigor of our manuscript.

2. In my previous review, I explicitly pointed out that setting all diagonal elements of the R matrix to 0.01 ppm is an overly simplistic and unrealistic treatment of observation error. I suggested that, at a minimum, the R matrix should reflect the variance of the observations at each site, which would introduce basic spatial heterogeneity and improve the physical realism of the assimilation system. However, in the authors' latest response, no changes were made to the R matrix design. The authors merely acknowledge that the 0.01 ppm setting is simplistic, but continue to use it without further justification or adjustment. Computing observation-based variances—even from synthetic data—is not technically difficult, especially in a twin experiment framework where the synthetic observations are fully defined. A more realistic R matrix would be straightforward to implement and would significantly improve the credibility of the assimilation framework. Therefore, I strongly urge the authors to either: Revise the R matrix to reflect spatially varying variances based on the observation time series, or Provide a quantitative justification (e.g., sensitivity analysis) showing that using a constant 0.01 ppm does not materially affect the assimilation results. Without such a revision or justification, the conclusions drawn from the assimilation experiments may not be robust or generalizable.

We thank you for your feedback regarding the R matrix in our manuscript. Your comments, particularly on the simplistic use of a uniform diagonal R matrix with a value of 0.01 ppm highlight an important aspect of data assimilation that warrants careful consideration.

We fully agree that, in applications involving real observations, the R matrix should incorporate at least spatial and, ideally, temporal variability to reflect observation error variances at different sites, thereby enhancing the physical realism of the assimilation system. In this study, our primary objective was to compare the performance of two data assimilation methods - VarDA and the EnVarDA - in solving

an inverse problem for parameter calibration within a complex atmospheric CO2 concentration modeling framework, specifically using the ORCHIDEE land surface model coupled with the LMDZ atmospheric transport model.

To focus on this methodological comparison, we employed a twin experiment framework with synthetic observations generated by ORCHIDEE+LMDZ without added noise or perturbations. This idealized setup eliminates complexities such as model structural errors or measurement errors, allowing the model to perfectly match the synthetic observations. In this context, we chose a uniform R matrix value of 0.01 ppm for both methods to ensure a consistent and controlled comparison, as the synthetic observations are noise-free and equally reliable across all stations. Assigning differential weights to stations via a heterogeneous R matrix seemed less relevant and sub-optimal in this perfect case, where all stations can theoretically be fitted equally well.

**We added L348:**

The R matrix represents the model-structural and observation errors. In our twin experiment setup, we choose not to add only very small errors in the pseudo-observations to compare both methods in an ideal case. In this context, structural errors in the ORCHIDEE LSM (i.e. missing processes, etc.) and in the transport model (i.e., coarse spatial resolution, wind biases, etc.), or measurement errors are discarded. Indeed, since the pseudo-observations are generated by a simulation, as detailed in Section 2.3, there exists at least one solution where all observations can be matched perfectly. For this reason, we use a simplified R matrix with the same small diagonal terms of 0.01 ppm for all stations. The rationale behind this choice is that, as all stations can be matched perfectly, we do not want to introduce any spatial or temporal preferences.

It is also clear that any change to the R matrix should change the results, as the opposite seems incorrect, but we believe that characterising the error in a system and testing the potential performance of a system are two different things. In our article, we do not wish to characterise the observation/model error (as there is none in reality), but we wish to test and compare two methods. We are convinced that as long as the same R is used in both systems, we can conclude that EnVarDA has many advantages over VarDA.

Moreover to assess the robustness of our approach and address the potential risk of overfitting due to the simplified R matrix, we evaluated the normalized chi-square statistic for the simple case, following Talagrand and Boutier (1999) and Trémolet (2006), obtaining scores of 0.015 for EnVarDA and 1.2 for VarDA. These values suggest that our error estimates are either appropriately specified or slightly

overestimated, as scores significantly greater than 1 would indicate error underestimation and a potential shift in the cost function minimum (Trémolet (2006)).

These findings support the validity of our conclusions within the controlled twin experiment framework. We recognize, however, that a more realistic R matrix incorporating spatially varying variances is critical for the use of real-observation, as we have observed in an ongoing work with real observations. We have also explicitly added the point raised by the reviewer in the discussion L565:

However, the assimilation of real observations is not straightforward. The use of real data must be followed by characterisation of the model/observation errors. Indeed, the matrix R must reflect modelling/observation errors at each site, which would introduce spatial heterogeneity, as each station may have different modelling errors, mainly structural errors from both the transport model and the ORCHIDEE flux model, or measurement problems. A good characterisation of the matrix R is of paramount importance, as it can have a considerable impact on the results obtained. If the model/observation errors are incorrect, the EnVarDA method can give infeasible posterior parameter values, i.e. outside the imposed parameter bound- aries (and therefore give non-physical parameter values). Furthermore, even with feasible posterior parameter values, the parameters obtained may be beyond the assumption of linearity made by the use of linear combinations in Eq. 18 and therefore do not improve the associated simulation. Nevertheless, several techniques seem promising for managing these limitations.

We believe this approach will address your concerns without necessitating a complete re-optimization of all experiments, which would be computationally intensive and time-consuming given the scope of the study.

3. The manuscript lacks a clear and detailed explanation of how the background error covariance matrix B is constructed. This is a fundamental component of any variational data assimilation framework, as it governs how prior uncertainty is propagated into the analysis. However, the manuscript appears to apply the same simplified approach to the background error covariance matrix B as it does to the observation error covariance matrix R—namely, by assigning constant diagonal values of 0.01. This practice is scientifically inappropriate. The background error covariance matrix and the observation error covariance matrix represent distinct sources of uncertainty and must be treated separately. Using the same constant value for both implicitly assumes that the model and observation uncertainties are identical in magnitude and structure, which is both unrealistic and unjustified—even in a twin experiment setup. Even

in idealized experiments, a scientifically grounded design of *B* is expected. For example, *B* could be derived from ensemble statistics, parameter perturbation experiments, or climatological variances. These are standard practices in both 3DVar and EnVar systems.

We thank the reviewer for their comment and apologise if our original explanation was unclear. We fully agree that the two matrices **R** and **B** must be treated separately and confirm that this is what we have done in the article. Note that we couldn't have done otherwise, because **R** relates to the error in atmospheric concentrations (in ppm), while **B** relates to the error in the model parameters, each of which has its own range of variation and unit. As discussed in the comment above **R** is indeed a diagonal matrix with an error of 0.01 ppm. But we clearly presented also in section 2.3.3 - how **B** is designed: "The **B** matrix contains the background errors associated with the prior knowledge of the model parameters. We set an error corresponding to 30% of the parameter range for the simple case and 20% for the complex case (as we use larger parameter ranges)." To improve clarity and avoid potential confusion, we have revised Section 2.3.3 to more explicitly separate the descriptions of **R** and **B** into two distinct paragraphs (see Lines 330–345 in the revised manuscript):

To implement the two data assimilation methods, ε-VarDA and EnVarDA, we define two error covariance matrices: R and B. These matrices are configured to be diagonal, as we are assimilating "synthetic" observations, and are common to both methods to ensure comparable experiments. Their configurations are informed by previous data assimilation studies using ORCHIDEE and a simplified carbon model (Kuppel et al., 2012, 2013; Bastrikov et al., 2018; MacBean et al., 2016), with Peylin et al. (2016) specifically applying diagonal matrices for atmospheric CO2 observations.

**R** Matrix

[...]

**B** Matrix

The B matrix represents the background errors associated with prior knowledge of the parameters. We set the error to 30% of the parameter range for the simple case and 20% for the complex case (as we use larger parameter ranges for this case). The background errors of each parameter can be seen in Fig. 3 for the simple case and in Fig. 6 as well as Table A2.

4. To improve clarity and transparency, I strongly recommend that the authors include a dedicated "Experiment Design" section in the manuscript, preferably early in the Methods section. Currently, the description of the different twin experiments is scattered and somewhat

difficult to follow, especially with regard to the distinctions between test cases, the naming conventions used, and the variables being optimized. Additionally, I suggest including a summary table that clearly outlines the different experiments conducted.

We thank the reviewer for these comments and apologise for the lack of clarity. We have modified the structure as follows:

**2.3 Experiment Design**

**2.3.1 Twin Experiment Description**

To test the data assimilation methods presented in Section 2.2, we conducted a so-called twin experiment to evaluate their efficiency in calibrating parameters involved in calculating NBP fluxes in the **ORCHIDEE LSM model. This experimental framework reduces** complexities associated with model-data errors, focusing on the performance of the assimilation methods. The known 'true' parameters being the default parameter values of the ORCHIDEE model are used to generate the synthetic observations. New values of a priori parameters are manually generated, ensuring physically meaningful values that differ from the 'true' parameters both presented in Table A2. The assimilation methods are then applied to assess how closely they converge toward the known solution (standard parameter values). The synthetic observations of atmospheric CO2 concentrations from the 21 continental stations are assimilated simultaneously over a two-year window (2000–2001) to monitor spatial and temporal variations in carbon fluxes, as shown in Figure A4. A limited period was chosen for practical reasons to avoid computationally expensive simulations.

**2.3.2 Generation of Synthetic Observations**

To generate synthetic observations for the twin experiment, we simulate net biome productivity (NBP) fluxes at the global scale using the ORCHIDEE LSM with default parameter values, referred to as the 'true' parameters (see Table A2). These NBP fluxes represent the net carbon fluxes of the land component, calculated as the difference between emission fluxes (heterotrophic and autotrophic respiration, and disturbance fluxes due to land-use change) and sink fluxes (primarily due to photosynthesis). The concentration given by the surface fluxes (the simulated NBP fluxes, along with other fluxes described in Section 2.1.4) are

transported using pre-calculated transport fields of the LMDZ model over the period 2000–2001. We then extract atmospheric CO2 concentrations at 21 continental atmospheric stations, shown in Figure 1, which are highly sensitive to continental carbon fluxes, providing significant constraints on the parameters. This process enabled the generation of synthetic observations of monthly average atmospheric CO2 concentrations at these 21 stations over the two-year period. It is important to note that the steps taken here to generate the synthetic observations are exactly the same as those used to perform a simulation. This means that there is at least one solution where the model can perfectly match the synthetic observation.

2.3.3 Simplified case

[...]

2.3.4 Complex case

[...]

**2.3.5** Error covariance matrices

[...]

**2.3.6** Tuning  $\epsilon$  for gradient calculation

[...]

**2.3.7** Defining the impact of the configuration

[...]

Each section was revised to reflect every aspect of the experiment we performed. We specifically split the first section into two parts to explicitly describe how the Synthetic Observations are generated, thereby removing any potential doubt. No additional information was added to the text but the text was revised to eliminate any potential confusion. We did include a list of all the experiments performed in the last section L380:

To assess their impact, we launch the twin experiment using different configurations:

for the simple case:

- $\circ$  5 different values of ε for the ε-VarDA based on the sensitivity test presented in Section 2.3.6;
- o 5 different ensemble sizes in the EnVarDA;
- for the complex case:
  - 5 different ensemble sizes in the EnVarDA;
  - 1 values of  $\epsilon$  for the  $\epsilon$ -VarDA.
- 5. In the author's response, it is stated that the experiments represent a full-field assimilation, implying that the assimilation directly updates the full state variables and can correct potential model biases. However, this characterization is not substantiated in the manuscript. There is no analysis or discussion demonstrating how the assimilation affects model biases—either in the prior fields, posterior fields, or fluxes. A full-field assimilation experiment should, by definition, lead to noticeable improvements in the state estimation compared to the biased model trajectory. To support this claim, I strongly recommend that the authors:

  1. Include an explicit evaluation of model biases before and after assimilation, especially in CO2 concentrations or fluxes (e.g., NBP); 2. Quantify the impact of assimilation on these biases.

Thank you for your feedback and for highlighting the need for a clear demonstration of how our assimilation approach addresses model biases in CO2 concentrations and fluxes, such as Net Biome Productivity (NBP).

In our study, we focus on calibrating parameters within the ORCHIDEE LSM that govern key processes such as photosynthesis and soil carbon decomposition, which in turn influence exchange fluxes like Gross Primary Production (GPP) and NBP. These parameters are optimized using atmospheric CO2 concentration data, while forcing variables (e.g., temperature, wind, precipitation) are prescribed from ERA-Interim reanalysis data to ensure accurate timing of meteorological events. As such, our assimilation does not directly update the state variables themselves but indirectly improves the model's representation of CO2 concentrations and fluxes by optimizing the parameters that drive these processes.

We acknowledge that the use of the term 'full-field assimilation' in our previous response could have been confusing. These terms are not used in the context of LSM parameter calibration (see, for example, the review articles by Raoult et al. 2024, Macbean et al. 2022, or Rayner et al. 2010).

To address your specific recommendations:

 Evaluation of Model Biases Before and After Assimilation: We believe that our manuscript already includes a comprehensive evaluation of biases in CO2 concentrations and fluxes. For CO2 concentrations, we quantify the improvement in model performance through the Root Mean Squared Difference (RMSD) scores, as presented in Figures 4 and 5, which compare prior and posterior simulations against synthetic observations at 21 atmospheric stations. Specifically, Figure 4 (and Figure A4 in the revised manuscript) shows the time series of CO2 concentrations, illustrating the reduction in discrepancies between model outputs and observations post-assimilation. For fluxes, Figure 7 and Figure A3 in the revised manuscript provide spatial differences in NBP and GPP fluxes, respectively, between prior and posterior estimates compared to "true" synthetic fluxes. These figures demonstrate the impact of assimilation on reducing biases in both CO2 concentrations and fluxes. We have also added a new analysis following Hodson et al. (2021) and Geman et al., 1992, who proposed to decompose the mean square difference (MSD) into bias and variance L451

We computed the mean squared difference (MSD) between the synthetic observations concatenated across all stations and the prior simulation, as well as the two posterior simulations. Following Hodson et al. (2021) and Geman et al. (1992), we decomposed the MSD into bias and variance terms as presented in Section A. The prior MSD is 11.49 ppm2 and is reduced to 0.04 ppm2 using the EnVarDA method and to 0.08 ppm2 using the VarDA method. The decomposition of the prior MSD indicates a squared bias of 4.96 ppm2 and an error variance equal to 6.53 ppm2. The same decomposition for the posterior simulations yields a squared bias of 0.006 ppm2 and an error variance equal to 0.03 ppm2 for the EnVarDA method, and a squared bias of 0.002 ppm2 and an error variance equal to 0.07 ppm2 for the VarDA method.

And L502

Furthermore the MSD score is better for the EnVarDA method (0.04 ppm2 using the EnVarDA method and to 0.08 ppm2 using the VarDA method) and the MSD decomposition (Geman et al., 1992; Hodson et al., 2021) highlights that EnVarDA better reduces the error variance, whereas the squared bias reduction is slightly better for the VarDA method. However, squared bias values below 0.01 ppm2 are negligible. While the mean RSMD reduction and MSD scores are similar for the complex case, the MAD scores in parameter space are different.

2. Quantification of Bias Impact: The manuscript quantifies the impact of assimilation on biases in CO2 concentrations and fluxes in the "Results" and "Discussion" sections. For CO2 concentrations, we report a mean RMSD reduction of 91.3% for 4DEnVar and 92.3% for e-4DVar across all stations (L 425–430, Page 17). For NBP and GPP, we discuss the recovery of global budgets and highlight challenges in achieving correct spatial distributions due to equifinality, particularly in 485–497 and 510–517 (Pages 21 and 23). These sections detail how both methods successfully reduce biases in global NBP and GPP budgets, although 4DEnVar performs better in capturing spatial patterns due to its ensemble-based approach, which mitigates issues related to local minima.

We believe these analyses directly address the impact of assimilation on model biases, as requested.

**References**

Asch, M., Bocquet, M., and Nodet, M.: Data Assimilation: Methods, Algorithms, and Applications, vol. 28, 2016.

Courtier, P., Thépaut, J., and Hollingsworth, A.: A strategy for operational implementation of 4D-Var, using an incremental approach, Quarterly Journal of the Royal Meteorological Society, 120, https://doi.org/10.1002/qj.49712051912, 1994.

Talagrand, O. and Courtier, P.: Variational Assimilation of Meteorological Observations With the Adjoint Vorticity Equation. I: Theory, Quarterly Journal of the Royal Meteorological Society, 113, <a href="https://doi.org/10.1002/gi.49711347812">https://doi.org/10.1002/gi.49711347812</a>, 1987.

Chevallier, F., Fisher, M., Peylin, P., Serrar, S., Bousquet, P., Bréon, F. M., Chédin, A., and Ciais, P.: Inferring CO2 sources and sinks from satellite observations: Method and application to TOVS data, Journal of Geophysical Research Atmospheres, 110, <a href="https://doi.org/10.1029/2005JD006390">https://doi.org/10.1029/2005JD006390</a>, 2005.

Basu, S., Guerlet, S., Butz, A., Houweling, S., Hasekamp, O., Aben, I., Krummel, P., Steele, P., Langenfelds, R., Torn, M., Biraud, S., Stephens, B., Andrews, A., and Worthy, D.: Global CO2 fluxes estimated from GOSAT retrievals of total column CO2, Atmospheric Chemistry and Physics, 13, <a href="https://doi.org/10.5194/acp-13-8695-2013">https://doi.org/10.5194/acp-13-8695-2013</a>, 2013.

Yannick Trémolet. 2007. Model-error estimation in 4D-Var. Quarterly Journal of the Royal Meteorological Society 133, 626, 1267–1280 <a href="https://doi.org/10.1002/qj.94">https://doi.org/10.1002/qj.94</a>, 2007.

O. Talagrand and F. Bouttier, Internal diagnostics of data assimilation systems, Conference Paper, <a href="https://www.ecmwf.int/en/elibrary/76594-internal-diagnostics-data-assimilation-systems">https://www.ecmwf.int/en/elibrary/76594-internal-diagnostics-data-assimilation-systems</a>, 1999.

Raoult, N., Douglas, N., MacBean, N., Kolassa, J., Quaife, T., Roberts, A. G., Fisher, R. A., Fer, I., Bacour, C., Dagon, K., Hawkins, L., Carvalhais, N., Cooper, E., Dietze, M., Gentine, P., Kaminski, T., Kennedy, D., Liddy, H. M., Moore, D., Peylin, P., Pinnington, E., Sanderson, B. M., Scholze, M., Seiler, C., Smallman, T. L., Vergopolan, N., Viskari, T., Williams, M., and Zobitz, J.: Parameter Estimation in Land Surface Models: Challenges and Opportunities with Data Assimilation and Machine Learning, ESS Open Archive, https://doi.org/10.22541/essoar.172838640.01153603/v1, 2024b.

MacBean, N., Bacour, C., Raoult, N., Bastrikov, V., Koffi, E. N., Kuppel, S., Maignan, F., Ottlé, C., Peaucelle, M., Santaren, D., and Peylin, P.: Quantifying and Reducing Uncertainty in Global Carbon Cycle Predictions: Lessons and Perspectives From 15 Years of Data Assimilation Studies With the ORCHIDEE Terrestrial Biosphere Model, Global Biogeochemical Cycles, 36, e2021GB007 177, <a href="https://doi.org/https://doi.org/10.1029/2021GB007177">https://doi.org/https://doi.org/10.1029/2021GB007177</a>, e2021GB007177, 2022.

Rayner, P. J.: The current state of carbon-cycle data assimilation, <a href="https://doi.org/10.1016/j.cosust.2010.05.005">https://doi.org/10.1016/j.cosust.2010.05.005</a>, 2010.

Bastrikov, V., Macbean, N., Bacour, C., Santaren, D., Kuppel, S., and Peylin, P.: Land surface model parameter optimisation using in situ flux data: Comparison of gradient-based versus random search algorithms (a case study using ORCHIDEE v1.9.5.2), Geoscientific Model Development, 11, <a href="https://doi.org/10.5194/gmd-11-4739-2018">https://doi.org/10.5194/gmd-11-4739-2018</a>, 2018.

Peylin, P., Bacour, C., MacBean, N., Leonard, S., Rayner, P., Kuppel, S., Koffi, E., Kane, A., Maignan, F., Chevallier, F., Ciais, P., and Prunet, P.: A new stepwise carbon cycle data assimilation system using multiple data streams to constrain the simulated land surface carbon cycle, Geoscientific Model Development, 9, <a href="https://doi.org/10.5194/gmd-9-3321-2016">https://doi.org/10.5194/gmd-9-3321-2016</a>, 2016.

Kuppel, S., Chevallier, F., and Peylin, P.: Quantifying the model structural error in carbon cycle data assimilation systems, Geoscientific Model Development, 6, <a href="https://doi.org/10.5194/qmd-6-45-2013">https://doi.org/10.5194/qmd-6-45-2013</a>, 2013.

Kuppel, S., Peylin, P., Chevallier, F., Bacour, C., Maignan, F., and Richardson, A. D.: Constraining a global ecosystem model with multi-site eddy-covariance data, Biogeosciences, 9, <a href="https://doi.org/10.5194/bg-9-3757-2012">https://doi.org/10.5194/bg-9-3757-2012</a>, 2012.

MacBean, N., Peylin, P., Chevallier, F., Scholze, M., and Schürmann, G.: Consistent assimilation of multiple data streams in a carbon cycle data assimilation system, Geoscientific Model Development, 9, <a href="https://doi.org/10.5194/gmd-9-3569-2016">https://doi.org/10.5194/gmd-9-3569-2016</a>, 2016.

Douglas, N., Quaife, T., and Bannister, R.: Exploring a hybrid ensemble–variational data assimilation technique (4DEnVar) with a simple ecosystem carbon model, Environmental Modelling & Software, 186, 106361, <a href="https://doi.org/10.1016/j.envsoft.2025.106361">https://doi.org/10.1016/j.envsoft.2025.106361</a>, 2025.

Pinnington, E., Quaife, T., Lawless, A., Williams, K., Arkebauer, T., and Scoby, D.: The Land Variational Ensemble Data Assimilation Framework: LAVENDAR v1.0.0, Geosci. Model Dev., 13, 55-69, 10.5194/gmd-13-55-2020, 2020.

Geman, S., Bienenstock, E., and Doursat, R.: Neural Networks and the Bias/Variance Dilemma, Neural Computation, 4, https://doi.org/10.1162/neco.1992.4.1.1, 1992.

Hodson, T. O., Over, T. M., and Foks, S. S.: Mean Squared Error, Deconstructed, Journal of Advances in Modeling Earth Systems, 13,

https://doi.org/10.1029/2021MS002681, 2021.